# Dr-Fairness: Dynamic Data Ratio Adjustment for Fair Training on Real and Generated Data

**Yuji Roh**[*]                                           *yuji.roh@kaist.ac.kr*
*KAIST*

**Weili Nie**                                             *wnie@nvidia.com*
*NVIDIA*

**De-An Huang**                                           *deahuang@nvidia.com*
*NVIDIA*

**Steven Euijong Whang**                                  *swhang@kaist.ac.kr*
*KAIST*

**Arash Vahdat**                                          *avahdat@nvidia.com*
*NVIDIA*

**Anima Anandkumar**                                      *aanandkumar@nvidia.com*
*NVIDIA*
*Caltech*

**Reviewed on OpenReview:** *https://openreview.net/forum?id=TyBd56VK7z*

## Abstract

Fair visual recognition has become critical for preventing demographic disparity. A major cause of model unfairness is the imbalanced representation of different groups in training data. Recently, several works aim to alleviate this issue using generated data. However, these approaches often use generated data to obtain similar amounts of data across groups, which is not optimal for achieving high fairness due to different learning difficulties and generated data qualities across groups. To address this issue, we propose a novel adaptive sampling approach that leverages both real and generated data for fairness. We design a bilevel optimization that finds the optimal data sampling ratios among groups and between real and generated data while training a model. The ratios are dynamically adjusted considering both the model's accuracy as well as its fairness. To efficiently solve our non-convex bilevel optimization, we propose a simple approximation to the solution given by the implicit function theorem. Extensive experiments show that our framework achieves state-of-the-art fairness and accuracy on the CelebA and ImageNet People Subtree datasets. We also observe that our method adaptively relies less on the generated data when it has poor quality. Our work shows the importance of using generated data together with real data for improving model fairness.

## 1 Introduction

Model fairness in visual recognition is becoming essential to prevent discriminatory predictions over demographics. Recently, numerous unfairness issues have been reported (Wang et al., 2020; Najibi, 2020), and several fair image classification approaches have been proposed that do not discriminate against specific groups such as gender, age, or skin color (Ramaswamy et al., 2021; Roh et al., 2021).

---

[*]Work done during an internship at NVIDIA.

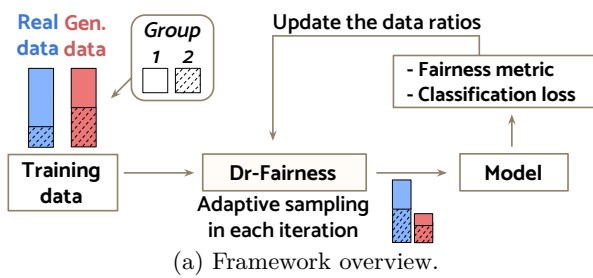

(a) Framework overview.

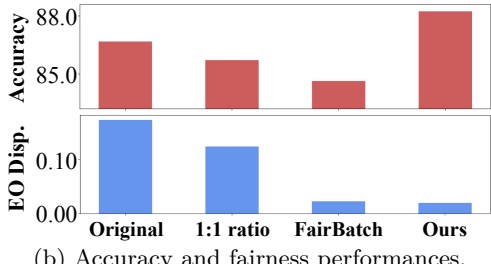

(b) Accuracy and fairness performances.

Figure 2: (a) Our framework iteratively updates the data ratios among groups and between real and generated data based on the fairness and accuracy of the intermediate model. (b) Performances on CelebA, using `gender` as the group attribute and `age` as the label attribute. Compared to the original model, the 1:1 ratio baseline (Ramaswamy et al., 2021) does not significantly improve group fairness, measured through equalized odds (EO) disparity. FairBatch (Roh et al., 2021) shows high fairness by adaptively selecting real data only, but loses accuracy. In comparison, Dr-Fairness (ours) achieves high fairness, while not sacrificing accuracy.

With the rapid progress in deep generative learning (Karras et al., 2020; Dhariwal & Nichol, 2021), there is a new research direction to improve fairness by augmenting training data with generated data. Recent breakthroughs in generative learning make generated data practical enough to use in real-world applications (OpenAI, 2022),

and many high-quality pre-trained generative models are now open to the public (Rombach et al., 2022), which obviates the need to retrain such models from scratch for new use cases. Thus, generated data is increasingly used to improve model performances, including fairness. From a fairness perspective, generated data complements real data by making it more diverse. For example, if a specific group's data is collected from a limited data source that does not have the full data distribution, that group may be discriminated in model training due to the bias (Mehrabi et al., 2021). In this case, generated data can be used to supplement that underrepresented group – see an example in Figure 1.

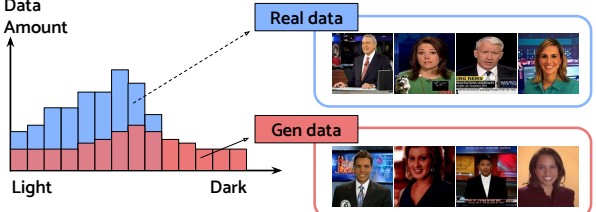

Figure 1: Many real-world image datasets have biased representation of protected groups (Wang et al., 2022). For example, ImageNet People Subtree (Yang et al., 2020) contains mostly lighter skin color images for anchors. Here, the generated data can compensate for the underrepresented groups, i.e., the darker skin color distributions for anchors.

However, most fair training approaches that use generated data simply generate similar amounts of data across groups (Ramaswamy et al., 2021; Choi et al., 2020), which may not be optimal to improve group fairness such as equalized odds (Hardt et al., 2016) and demographic parity (Feldman et al., 2015). Such suboptimality could originate from 1) the learning difficulty differences across groups and 2) the potential bias (i.e., typically in the form of missing modes) issues in the generated data that can hurt the fairness of the model under training. Moreover, as the quality of the generated data can differ among different group groups, simply using a similar amount of generated data may result in suboptimal model accuracy and fairness. Therefore, it is essential to find the right mix of generated and real data for the best accuracy and fairness.

In this paper, we *harness the potential of both real and generated data via adaptive sampling* to improve group fairness while minimizing accuracy degradation. To this end, we design a new sampling approach called Dr-Fairness (**D**ynamic Data **R**atio Adjustment for **Fairness**) that adaptively adjusts data ratios among groups and between real and generated data over iterations, as in Figure 2a.

In Table 1, we compare the unique properties of Dr-Fairness against two representative methods: 1) an equal ratio baseline (1:1 ratio) (Ramaswamy et al., 2021) that uses generated data and 2) a fairness-aware adaptive sampling baseline (FairBatch) (Roh et al., 2021) that finds the optimal group ratio for fairness only using real data. We can see that Dr-Fairness subsumes the two baselines and improves them by also optimizing the ratio between real and generated data and utilizing accuracy for ratio updates.

Table 1: Functionality comparison of algorithms.

| Method | Uses generated data | Finds the optimal group ratio | Finds the optimal real & gen. data ratio | Utilizes accuracy for ratio updates |
|---|---|---|---|---|
| 1:1 ratio | ✓ | ✗ | ✗ | ✗ |
| FairBatch | ✗ | ✓ | ✗ | ✗ |
| **Dr-Fairness** | ✓ | ✓ | ✓ | ✓ |

To perform adaptive sampling systematically, we design a novel bilevel optimization problem along with an efficient algorithm for solving it. Our bilevel optimization consists of 1) an outer optimization that adjusts data sampling ratios considering both fairness and accuracy and 2) an inner optimization that minimizes the standard empirical risk on both real and generated data, given the current sampling ratios. Although various exact algorithms have been proposed to solve bilevel optimizations (Maclaurin et al., 2015), they often scale poorly in our scenario with large models and data. We thus propose an approximate algorithm that uses the implicit function theorem (Krantz & Parks, 2002) and identity-matrix approximation (Luketina et al., 2016) to efficiently compute the gradient of our bilevel optimization. Specifically, instead of computing the expensive inverse Hessian matrix, we approximate it with a simple diagonal identity matrix.

Experiments on CelebA (Liu et al., 2015) and ImageNet People Subtree (Yang et al., 2020) show that our approach achieves the state-of-the-art fairness and accuracy performances. For instance, Figure 2b highlights our results on CelebA, where our framework largely outperforms FairBatch, which only uses real data and the 1:1 ratio baseline – see Sec. 4 for comparisons using more baselines and other fairness metrics, which show consistent results. On the ImageNet People Subtree classification problem, which represents a large-scale real-world scenario, we achieve better accuracies than the best baseline, with an absolute improvement of 5–9%, while obtaining similar fairness scores. We also observe that our framework adaptively relies less on the generated data when it has poor quality.

**Summary of Contributions:** (1) We propose Dr-Fairness, a novel adaptive sampling framework for fair training that enjoys the potential of both real and generated data. (2) To perform adaptive sampling systematically, we formulate a bilevel optimization to train fair and accurate models on real and generated data. (3) We also design an approximate algorithm based on the implicit function theorem and identity-matrix approximation to efficiently solve our non-convex optimization. (4) We perform extensive experiments on CelebA and ImageNet People Subtree to show that Dr-Fairness achieves the state-of-the-art accuracy and fairness. (5) Finally, our work reveals the importance of using generated data together with real data to improve model fairness.

## 2   Related Work

**Traditional Model Fairness**   As model fairness becomes indispensable for Trustworthy AI, numerous works have been recently proposed to better measure fairness and design fairness-aware algorithms (Narayanan, 2018). Among various fairness definitions, we focus on group fairness measures (Hardt et al., 2016; Feldman et al., 2015), which are widely studied in the fairness literature. The main approaches for satisfying group fairness are: 1) fix the training data to mitigate bias (Kamiran & Calders, 2011; Zemel et al., 2013), 2) modify the training process to prevent the model from learning bias (Zafar et al., 2017a;b; Zhang et al., 2018a; Agarwal et al., 2018; Roh et al., 2020; 2021), or 3) alter the outputs of the trained model to achieve fairness metrics (Hardt et al., 2016). Unfortunately most of algorithms are not designed to handle large number of groups or labels, and our contribution is to support such large-scale scenarios for real-world applications. There are several recent works that aim to improve fairness for multiple numbers of groups (Shui et al., 2022b; Zhao et al., 2021), but they only focus on specific fairness definitions (Shui et al., 2022b), or the performance improvement is limited (Zhao et al., 2021) compared to our work.

Among the previous techniques, FairBatch (Roh et al., 2021) is the most relevant to our work as it finds the optimal group ratio for fairness on real data and shows the state-of-the-art fairness performances on various tabular datasets, including COMPAS (Angwin et al., 2016) and AdultIncome (Kohavi, 1996). However, FairBatch may suffer from accuracy degradation due to oversampling on very small-size groups, especially in vision datasets. In particular, FairBatch aims to optimize only the fairness criterion in the absence of any generated data, and it cannot be easily extended to optimize both accuracy and fairness objectives together, nor can it utilize generated data. Also, the theoretical guarantees of FairBatch do not apply in our non-binary setting because the objectives of our optimization problem become non-convex – details on the optimization are in Sec. 3. In contrast, Dr-Fairness can minimize the accuracy degradation of fair training by optimally utilizing both real and generated data based on the fairness and accuracy objectives.

In addition, there are other related studies on fair data reweighing (Li & Liu, 2022; Jiang & Nachum, 2020; Krasanakis et al., 2018), fair augmentation (Chuang & Mroueh, 2021), and fair representations (Shui et al.,

2022a). Compared to our work, these studies only use real data or do not scale to large datasets. For example, applying the existing fair reweighing techniques on large-scale data may lead to significant training times due to multiple re-trainings (Jiang & Nachum, 2020; Krasanakis et al., 2018) or performance degradation due to violations of the underlying assumptions (Li & Liu, 2022). We leave a detailed discussion in Sec. C.

**Fairness in Visual Recognition**   There is an emerging line of research for fairness in visual recognition (Najibi, 2020; Wang et al., 2020) where using generated data is critical. Many visual recognition tasks involve multiple classes of varying sizes, and only using real data is often insufficient to improve fairness. In response, several works have proposed new algorithms to create a balanced dataset by augmenting the biased real dataset with well-controlled generated data (Sattigeri et al., 2019; Choi et al., 2020; Ramaswamy et al., 2021). However, simply balancing the data sizes is not enough to achieve high-enough group fairness, as the learning difficulty and generated data quality can differ across groups. Although a recent work (Zietlow et al., 2022) suggests an adaptive data augmentation that generating more data for worse-performing groups, it uses heuristics to adjust data ratios without proper optimization and thus has limited fairness performance. In comparison, Dr-Fairness solves a novel optimization problem to find optimal data ratios and thus obtains both high fairness and accuracy.

**Other Related Work**   Although not our immediate focus, there are other important research lines for fairness: 1) fulfilling other fairness definitions (e.g., individual fairness (Dwork et al., 2012) and causal fairness (Kusner et al., 2017)), 2) handling noisy or missing group labels (Hashimoto et al., 2018; Celis et al., 2021), and 3) improving fairness in special classification scenarios (e.g., selective classification (Lee et al., 2021)). We believe that supporting these aspects can be promising future directions.

## 3   Framework

In this section, we first formulate a bilevel optimization problem for optimizing sampling ratios for real and generated data. We then design a new algorithm that efficiently solves the optimization problem. Throughout this paper, we use the following notations and fairness definitions.

**Notations**   Let $x \in \mathbb{X}$ be the input feature, and let $y \in \mathbb{Y}$ and $\hat{y} \in \mathbb{Y}$ be the true label and the predicted label, respectively. Let $z \in \mathbb{Z}$ be a sensitive group attribute, e.g., gender, age, or skin color. Let $m$ be the total number of training samples, and $m_{y,z}$ be the number of samples in the set $\{i : y_i = y, z_i = z\}$ with label y and group label z. Similarly, $m_{y,\star} := |\{i : y_i = y\}|$ and $m_{\star,z} := |\{i : z_i = z\}|$. Let $\boldsymbol{w}$ be the model weights, and the overall empirical risk is given by $L(\boldsymbol{w}) = \frac{1}{m} \sum_i \ell(y_i, \hat{y}_i)$, where $\ell(\cdot)$ represents the loss function. Let $L_{y,z}(\boldsymbol{w})$ be the empirical risk over samples in the set $\{i : y_i = y, z_i = z\}$, i.e., $L_{y,z}(\boldsymbol{w}) := \frac{1}{m_{y,z}} \sum_{i:y_i=y,z_i=z} \ell(y_i, \hat{y}_i)$. Finally, let $L^{\mathrm{real}}(\cdot)$ and $L^{\mathrm{gen}}(\cdot)$ be the empirical risks on real data and generated data, respectively.

**Fairness Definitions**   For the method design, we focus on two prominent group fairness definitions: equalized odds (EO) (Hardt et al., 2016) and demographic parity (DP) (Feldman et al., 2015). EO is satisfied when the accuracies conditioned on the true label are the same for different groups (i.e., $\Pr(\hat{y} = y | y = y, z = z_1) = \Pr(\hat{y} = y | y = y, z = z_2), \forall y \in \mathbb{Y}, z_1, z_2 \in \mathbb{Z}$). DP is satisfied when the positive prediction rates are the same for the groups (i.e., $\Pr(\hat{y} = 1 | z = z_1) = \Pr(\hat{y} = 1 | z = z_2), \forall z_1, z_2 \in \mathbb{Z}$), where DP is designed for binary classifications (i.e., $y \in \{0, 1\}$) with a favorable label class (e.g., "approval" in loan decision). We show how to measure the disparities (i.e., unfairness) among groups for each fairness definition in Sec. 4.

**Generated Data**   In general, any synthetic data, including data from deep generative models, can be considered generated data for fair training. Here, the key role of the generated data in algorithmic fairness is supporting the limited subset of the real data. Also, we implicitly assume that the domains of the generated and real data are the same, but the distributions of the two data can be different. For example, if the real data represents human faces, then we assume the generated data also contains human faces. However, generated data may have a fairer distribution than real data. In this paper, we assume we can get group-specific generated data by using conditional image generation techniques (Nie et al., 2021; Dhariwal & Nichol, 2021) – see details in Secs. 4 and B.3.

### 3.1 Bilevel Optimization for Fairness with Real and Generated Data

To design an adaptive sampling strategy on real and generated data, we first formulate a bilevel optimization for training fair and accurate models. The bilevel optimization consists of inner and outer objectives: 1) we maintain the standard empirical risk minimization (ERM) in the inner problem, and 2) we capture the desired fairness properties in the outer problem. The bilevel formulation allows us to support prominent group fairness metrics and utilize generated data for fairness.

We now explain how our optimization improves group fairness and accuracy together by using both real and generated data. The outer objective aims to find the optimal data ratios among sensitive groups and between real and generated data to minimize the fairness and accuracy losses on the real data distribution. Given the current data ratios, the inner objective runs a weighted ERM with both real and generated data. We can support various prominent group fairness metrics by modifying the outer objective and the constraints. As an illustration, we state our bilevel optimization w.r.t. EO as follows (see the DP version in Sec. A.1):

$$\min_{\boldsymbol{\lambda},\boldsymbol{\mu}} \max_{y\in\mathbb{Y}, z_1,z_2\in\mathbb{Z}}\{|L^{\mathrm{real}}_{y,z_1}(\boldsymbol{w}(\boldsymbol{\lambda},\boldsymbol{\mu})) - L^{\mathrm{real}}_{y,z_2}(\boldsymbol{w}(\boldsymbol{\lambda},\boldsymbol{\mu}))|\} + k\sum_{y\in\mathbb{Y}, z\in\mathbb{Z}}\frac{m_{y,z}}{m}L^{\mathrm{real}}_{y,z}(\boldsymbol{w}(\boldsymbol{\lambda},\boldsymbol{\mu})),$$

$$\boldsymbol{w}(\boldsymbol{\lambda},\boldsymbol{\mu}) = \arg\min_{\boldsymbol{w}}\sum_{y\in\mathbb{Y}, z\in\mathbb{Z}}\frac{m_{y,\star}}{m}\lambda_{y,z}\{\mu_{y,z}L^{real}_{y,z}(\boldsymbol{w}) + (1-\mu_{y,z})L^{\mathrm{gen}}_{y,z}(\boldsymbol{w})\},$$

$$\text{s.t.} \quad \boldsymbol{\lambda}\in[0,1], \boldsymbol{\mu}\in[0,1], \sum_{z\in\mathbb{Z}}\lambda_{y,z}=1, \forall y\in\mathbb{Y},$$

where $\lambda_{y,z}$ is the ratio for group $z$ in class $y$, $\mu_{y,z}$ is the ratio for real data in the $(y,z)$-class, and $\boldsymbol{\lambda}$ and $\boldsymbol{\mu}$ are the sets of all $\lambda_{y,z}$ and $\mu_{y,z}$, respectively. In the outer objective, the first term indicates the fairness loss, and second term indicates accuracy loss. The hyperparameter $k$ tunes the importance of the two losses. We note that the $\lambda_{y,z}$ and $\mu_{y,z}$ values are the data ratios within one mini-batch. Thus, among all samples in the real and generated data, our framework serves mini-batches according to $\lambda_{y,z}$ and $\mu_{y,z}$. Here, we can capture the EO disparity as the maximum of the loss differences in different groups within the same label (i.e., $\max|L^{\mathrm{real}}_{y,z_1}(\boldsymbol{w}) - L^{\mathrm{real}}_{y,z_2}(\boldsymbol{w})|$).

**Remark 1.** *We explain how this loss-based constraint can capture equalized odds. When the loss function $\ell(\mathrm{y}_i,\hat{\mathrm{y}}_i)$ is 1/0-loss (i.e., $\ell(\mathrm{y}_i,\hat{\mathrm{y}}_i) = 1(\mathrm{y}_i \neq \hat{\mathrm{y}}_i)$, where $1(\cdot)$ is an indicator function), the loss-based constraint can perfectly express the equalized odds disparity. Specifically, $L_{y,z}(\boldsymbol{w})$ with 1/0-loss is equivalent to the probability of the correct predictions in each $(y, z)$-class (i.e., $\Pr(\hat{\mathrm{y}} = y|\mathrm{y} = y, \mathrm{z} = z)$). Therefore, our fairness loss constraint (i.e., $\max_{y\in\mathbb{Y}, z_1,z_2\in\mathbb{Z}}\{|L^{real}_{y,z_1}(\boldsymbol{w}(\boldsymbol{\lambda},\boldsymbol{\mu})) - L^{real}_{y,z_2}(\boldsymbol{w}(\boldsymbol{\lambda},\boldsymbol{\mu}))|\}$) becomes the equalized odds metric, which describes the class-conditioned accuracy disparity among groups (i.e., $\max_{y\in\mathbb{Y}, z_1,z_2\in\mathbb{Z}}|\Pr(\hat{\mathrm{y}} = y|\mathrm{y} = y, \mathrm{z} = z_1) - \Pr(\hat{\mathrm{y}} = y|\mathrm{y} = y, \mathrm{z} = z_2)|$). In practice, we can also use other loss functions like cross-entropy loss instead of the 1/0-loss, as other loss functions have been empirically verified as reasonable proxies for capturing group fairness metrics (Roh et al., 2021; Shen et al., 2022).*

Through the above formulation, the amount of generated data is automatically adjusted to augment the real data (e.g., enhancing minority groups in the real data). In particular, proper usage of generated data can reduce the accuracy degradation caused by over-sampling minority groups from real data.

**Advantages of Using Bilevel Formulation**   The bilevel formulation has various advantages in solving our problem. First, we can achieve the desired fairness properties while keeping the standard model training process without re-configuring the model architecture or loss functions. Moreover, our bilevel problem can be solved via an efficient algorithm that we propose in the following section, which is suitable to support a large number of groups and label classes. We note that when the numbers of groups and label classes increase (i.e., the numbers of $\lambda_{y,z}$ and $\mu_{y,z}$ increase), naive formulations like grid search using the validation set may fail to find reasonable solutions within a practical time. In Sec. A.5, we discuss more advantages of using bilevel optimization compared to other problem formulation methods like distributionally robust optimization (Sinha et al., 2017).

### 3.2 Algorithm

We now design our algorithm to solve the above bilevel optimization. In this section, we first describe how to efficiently approximate our optimization by utilizing the implicit function theorem (Krantz & Parks, 2002)

and adapting identity-matrix approximation (Luketina et al., 2016) in a fairness setting. We then introduce the overall training procedure, and show the validity of our approximate algorithm on synthetic data.

**Algorithm Design** Solving bilevel optimization is known to be challenging (Liu et al., 2021), especially when the objectives are non-convex as in our problem. Thus, we resort to stochastic gradient descent to find the optimal parameters of the bilevel optimization gradually. To obtain the gradients, we first convert our optimization into the unconstrained version:

$$\min_{\boldsymbol{\lambda},\boldsymbol{\mu}} \max_{y\in\mathbb{Y},z_1,z_2\in\mathbb{Z}} \{|L_{y,z_1}^{\mathrm{real}}(\boldsymbol{w}(\boldsymbol{\lambda},\boldsymbol{\mu})) - L_{y,z_2}^{\mathrm{real}}(\boldsymbol{w}(\boldsymbol{\lambda},\boldsymbol{\mu}))|\} + k \sum_{y\in\mathbb{Y},z\in\mathbb{Z}} \frac{m_{y,z}}{m} L_{y,z}^{\mathrm{real}}(\boldsymbol{w}(\boldsymbol{\lambda},\boldsymbol{\mu})),$$

$$\boldsymbol{w}(\boldsymbol{\lambda},\boldsymbol{\mu}) = \arg\min_{\boldsymbol{w}} \sum_{y\in\mathbb{Y},z\in\mathbb{Z}} \frac{m_{y,\star}}{m} \sigma_y(\lambda_{y,z})\{S(\mu_{y,z})L_{y,z}^{\mathrm{real}}(\boldsymbol{w}) + (1 - S(\mu_{y,z}))L_{y,z}^{\mathrm{gen}}(\boldsymbol{w})\},$$

where $\sigma_y(\lambda_{y,z}) := \exp(\lambda_{y,z})/\sum_{z_i}\exp(\lambda_{y,z_i})$ (i.e., the softmax function), and $S(\mu_{y,z}) := 1/(1 + \exp(-\mu_{y,z}))$ (i.e., the sigmoid function). Denoting the outer and inner objectives as $f_{\mathrm{outer}}(\boldsymbol{\lambda},\boldsymbol{\mu},\boldsymbol{w}(\boldsymbol{\lambda},\boldsymbol{\mu}))$ and $f_{\mathrm{inner}}(\boldsymbol{\lambda},\boldsymbol{\mu},\boldsymbol{w})$ respectively, the inner optimization $\boldsymbol{w}(\boldsymbol{\lambda},\boldsymbol{\mu}) = \arg\min_{\boldsymbol{w}} f_{\mathrm{inner}}(\boldsymbol{\lambda},\boldsymbol{\mu},\boldsymbol{w})$ can be solved efficiently using SGD-like algorithms. The main question is how to solve the outer optimization. We can state the gradient of $f_{\mathrm{outer}}$ w.r.t. $\boldsymbol{\lambda}$ as follows:

$$\frac{df_{\mathrm{outer}}}{d\boldsymbol{\lambda}} = \underbrace{\frac{\partial f_{\mathrm{outer}}}{\partial\boldsymbol{\lambda}}}_{\text{Term A}} + \underbrace{\frac{\partial f_{\mathrm{outer}}}{\partial\boldsymbol{w}(\boldsymbol{\lambda},\boldsymbol{\mu})}}_{\text{Term B}} \times \underbrace{\frac{\partial\boldsymbol{w}(\boldsymbol{\lambda},\boldsymbol{\mu})}{\partial\boldsymbol{\lambda}}}_{\text{Term C}}, \tag{1}$$

where Term A and Term B are the direct gradients w.r.t. $\boldsymbol{\lambda}$ and $\boldsymbol{w}(\boldsymbol{\lambda},\boldsymbol{\mu})$, respectively, and Term C is the best-response Jacobian. Note that $\boldsymbol{w}(\boldsymbol{\lambda},\boldsymbol{\mu})$ is the best-response of model weights.

In Eq. 1, the best-response Jacobian is hard to directly compute. Although various algorithms have been proposed to *explicitly* find the best-response Jacobian, most of them require propagating the entire history of the gradients (Maclaurin et al., 2015), which is very time-consuming. Instead, we *implicitly* measure the best-response Jacobian using the implicit function theorem (Krantz & Parks, 2002). This approach does not need to investigate the entire gradient history (Rajeswaran et al., 2019; Lorraine et al., 2020) and builds on the assumption that the inner optimization has converged to a local minimum, i.e., $\frac{\partial f_{\mathrm{inner}}}{\partial\boldsymbol{w}} = 0$. We note that among various methods of solving bilevel optimization problems, the implicit function theorem significantly improves the algorithm efficiency with theoretical evidence.

We now describe the details on how we convert the best-response Jacobian in Eq. 1 using the implicit function theorem. Here, we first state the original implicit function theorem:

**Theorem 2.** *(Implicit Function Theorem, stated in Krantz & Parks (2002); de Oliveira (2014)) Let $F : \mathbb{R}^n \times \mathbb{R}^m \to \mathbb{R}^m$ be a continuously differentiable function, where the input of $F$ is $(\boldsymbol{x}, \boldsymbol{y}) \in \mathbb{R}^n \times \mathbb{R}^m$. Assume there is an input point $(\boldsymbol{a},\boldsymbol{b})$ that satisfies $F(\boldsymbol{a},\boldsymbol{b}) = \boldsymbol{0}$, and $\frac{\partial F(\boldsymbol{a},\boldsymbol{b})}{\partial\boldsymbol{y}}$ (i.e., the Jacobian matrix) is invertible. Then, there exist open sets $U \subset \mathbb{R}^n$ and $V \subset \mathbb{R}^m$ that contain $\boldsymbol{a}$ and $\boldsymbol{b}$, respectively, and satisfy the following:*

- *There is a unique continuously differentiable function $G$, where $G(\boldsymbol{a}) = \boldsymbol{b}$ and $F(\boldsymbol{x}, G(\boldsymbol{x})) = \boldsymbol{0}$ for all $\boldsymbol{x} \in U$.*

- *We have the Jacobian matrix of partial derivatives of $G$ in $U$ as follows:*
$$\frac{\partial G(\boldsymbol{x})}{\partial\boldsymbol{x}} = -[\frac{\partial F(\boldsymbol{x}, G(\boldsymbol{x}))}{\partial\boldsymbol{y}}]^{-1}[\frac{\partial F(\boldsymbol{x}, G(\boldsymbol{x}))}{\partial\boldsymbol{x}}].$$

We now apply the above theorem in our setting. To get the best-response Jacobian w.r.t. $\boldsymbol{\lambda}$, we consider $\frac{\partial f_{\mathrm{inner}}(\boldsymbol{\lambda},\boldsymbol{w})}{\partial\boldsymbol{w}}$ as $F(\boldsymbol{x},\boldsymbol{y})$ and $\boldsymbol{w}(\boldsymbol{\lambda})$ as $G(\boldsymbol{x})$. Note that when accessing the gradient w.r.t. $\boldsymbol{\lambda}$, we can ignore $\boldsymbol{\mu}$ without loss of generality, and vice versa. Then, we can rewrite Theorem 2 for our scenario as follows:

**Corollary 3.** *(Implicit Function Theorem in our setting) Let $\frac{\partial f_{inner}}{\partial\boldsymbol{w}} : \mathbb{R}^n \times \mathbb{R}^m \to \mathbb{R}^m$ be a continuously differentiable function, where the input of $\frac{\partial f_{inner}}{\partial\boldsymbol{w}}$ is $(\boldsymbol{\lambda},\boldsymbol{w}) \in \mathbb{R}^n \times \mathbb{R}^m$. Assume there is an input point $(\boldsymbol{a},\boldsymbol{b})$ that satisfies $\frac{\partial f_{inner}(\boldsymbol{a},\boldsymbol{b})}{\partial\boldsymbol{w}} = \boldsymbol{0}$, and $\frac{\partial^2 f_{inner}(\boldsymbol{a},\boldsymbol{b})}{\partial\boldsymbol{w}\partial\boldsymbol{w}}$ (i.e., the Jacobian matrix) is invertible. Then, we have the Jacobian matrix of partial derivatives of $\boldsymbol{w}(\boldsymbol{\lambda})$ as follows:*
$$\frac{\partial\boldsymbol{w}(\boldsymbol{\lambda})}{\partial\boldsymbol{\lambda}} = -[\frac{\partial^2 f_{inner}(\boldsymbol{\lambda},\boldsymbol{w})}{\partial\boldsymbol{w}\partial\boldsymbol{w}}]^{-1} \times \frac{\partial^2 f_{inner}(\boldsymbol{\lambda},\boldsymbol{w})}{\partial\boldsymbol{w}\partial\boldsymbol{\lambda}}.$$

| **Algorithm 1:** Model Training with Dr-Fairness | **Algorithm 2:** Dr-Fairness |
|---|---|
| **Input:** real data $(x_{\text{real}}, y_{\text{real}}, z_{\text{real}})$, generated data $(x_{\text{gen}}, y_{\text{gen}}, z_{\text{real}})$ | **Input:** model parameters $\boldsymbol{w}$, data $\boldsymbol{d}_{\text{real}}$ and $\boldsymbol{d}_{\text{gen}}$, group ratio $\boldsymbol{\lambda}$, real data ratio $\boldsymbol{\mu}$ |
| $\boldsymbol{d}_{\text{real}}, \boldsymbol{d}_{\text{gen}} \leftarrow (x_{\text{train}}, y_{\text{train}}, z_{\text{real}}), (x_{\text{gen}}, y_{\text{gen}}, z_{\text{real}})$ | Calculate $f_{\text{outer}}$ and $f_{\text{inner}}$ according to $\boldsymbol{w}$, $\boldsymbol{d}_{\text{real}}, \boldsymbol{d}_{\text{gen}}, \boldsymbol{\lambda}$, and $\boldsymbol{\mu}$ |
| $\boldsymbol{w} \leftarrow$ initial model parameters | |
| $\boldsymbol{\lambda}, \boldsymbol{\mu} \leftarrow$ initialize sampling ratio logits | Get $\frac{df_{\text{outer}}}{d\boldsymbol{\lambda}}$ and $\frac{df_{\text{outer}}}{d\boldsymbol{\mu}}$ as in Eq. 3 |
| Get current sampling ratios $\sigma_y(\boldsymbol{\lambda}), S(\boldsymbol{\mu})$ | Update $\boldsymbol{\lambda}$ by $\frac{df_{\text{outer}}}{d\boldsymbol{\lambda}}$ and $\boldsymbol{\mu}$ by $\frac{df_{\text{outer}}}{d\boldsymbol{\mu}}$ using |
| **for** *each iteration* **do** | optimizers (e.g., Adam) |
|   minibatch=Dr-Fairness$(\boldsymbol{w}, \boldsymbol{d}_{\text{real}}, \boldsymbol{d}_{\text{gen}}, \boldsymbol{\lambda}, \boldsymbol{\mu})$ | Draw a minibatch w.r.t. $\sigma_y(\boldsymbol{\lambda})$, $S(\boldsymbol{\mu})$ |
|   Update $\boldsymbol{w}$ according to the minibatch (optionally with exponential moving average (EMA)) | |
| **Output:** model parameters $\boldsymbol{w}$ | **Output:** minibatch |

Thus, with the assumption that the inner optimization has converged to a local minimum, i.e., $\frac{\partial f_{\text{inner}}}{\partial \boldsymbol{w}} = 0$, we can convert Eq. 1 into the following equation by replacing the best-response Jacobian to the multiplication of two matrices:

$$\frac{df_{\text{outer}}}{d\boldsymbol{\lambda}} = \frac{\partial f_{\text{outer}}}{\partial \boldsymbol{\lambda}} + \frac{\partial f_{\text{outer}}}{\partial \boldsymbol{w}(\boldsymbol{\lambda}, \boldsymbol{\mu})} \times -\left[\frac{\partial^2 f_{\text{inner}}}{\partial \boldsymbol{w} \partial \boldsymbol{w}}\right]^{-1} \times \frac{\partial^2 f_{\text{inner}}}{\partial \boldsymbol{w} \partial \boldsymbol{\lambda}}. \tag{2}$$

However, obtaining the inverse Hessian (i.e., $[\partial^2 f_{\text{inner}}/\partial \boldsymbol{w} \partial \boldsymbol{w}]^{-1}$) in Eq. 2 is also computationally expensive. Thus, we consider the identity matrix approximation (Luketina et al., 2016; Finn et al., 2017; Geng et al., 2021) that replaces the inverse Hessian with the identity matrix. Despite its simplicity, such approximation may be valid for neural networks with normalization layers that make the Hessian matrix diagonally dominant (e.g. BatchNorm), and in practice, it often performs on par with other approximation methods in various applications (Raiko et al., 2012; Pedregosa, 2016; Liu et al., 2018; Wilder et al., 2019; Fung et al., 2022). Given this, we can rewrite Eq. 2 simply as:

$$\frac{df_{\text{outer}}}{d\boldsymbol{\lambda}} \approx \frac{\partial f_{\text{outer}}}{\partial \boldsymbol{\lambda}} - \frac{\partial f_{\text{outer}}}{\partial \boldsymbol{w}(\boldsymbol{\lambda}, \boldsymbol{\mu})} \times \frac{\partial^2 f_{\text{inner}}}{\partial \boldsymbol{w} \partial \boldsymbol{\lambda}}, \tag{3}$$

where the second term on the right-hand side is efficiently computed via vector-Jacobian product (Paszke et al., 2017). Similarly, we can also approximate the gradient of $f_{\text{outer}}$ w.r.t. $\boldsymbol{\mu}$.

**Remark 4.** *In Eq. 3, the second term dominates the computation cost as it performs both the Jacobian computation and vector-Jacobian product. In comparison, the computation cost of the first term is negligible. Here the computational cost increases when the model parameter size (i.e., dimension of $\boldsymbol{w}$) and the number of groups and label classes (i.e., dimension of $\boldsymbol{\lambda}$ and $\boldsymbol{\mu}$) increase. In our experiments, we update $\boldsymbol{\lambda}$ and $\boldsymbol{\mu}$ at every iteration of the training, but we can further reduce the computational cost by updating $\boldsymbol{\lambda}$ and $\boldsymbol{\mu}$ less frequently – see an empirical analysis for computational overhead in Sec. A.2.*

**Overall Training Process** We now describe the overall training process in Algo. 1. We first initialize the model parameters and the data ratios, and for each iteration, we then get a minibatch from Dr-Fairness (Algo. 2). Here, we note that Eq. 3 computes the gradients of ratios (i.e., $\boldsymbol{\lambda}$ and $\boldsymbol{\mu}$), rather than obtaining an analytical solution for them, thus we need to specify initial ratios in the training. In Algo. 2, we first update the data ratios among groups ($\boldsymbol{\lambda}$) and between real and generated data ($\boldsymbol{\mu}$) by calculating $\frac{df_{\text{outer}}}{d\boldsymbol{\lambda}}$ and $\frac{df_{\text{outer}}}{d\boldsymbol{\mu}}$ as in Eq. 3. We then draw a minibatch according to $\sigma_y(\boldsymbol{\lambda})$ and $S(\boldsymbol{\mu})$. Note that the batch sampling with $\sigma_y(\boldsymbol{\lambda})$ and $S(\boldsymbol{\mu})$ provides an unbiased estimator of the weighted ERM in our inner optimization (Roh et al., 2021). Finally, we update the model parameters $\boldsymbol{w}$ with the given minibatch. Here we can optionally use an exponential moving average (EMA) that averages the model parameters $\boldsymbol{w}$ for improving training stability.

**Validity of Our Algorithm** We empirically verify how close the solutions from our approximation strategy are to the optimal ones. To this end, we follow the synthetic binary setting in Roh et al. (2021), where FairBatch has a theoretical guarantee to find the optimal group ratios, and compare the optimized group ratios of Dr-Fairness and FairBatch – see details on the setup in Sec. A.3. Note that in this synthetic setting, we set the fairness metric to equal opportunity (i.e., a relaxed version of EO that focuses on the positive label) and only use the real data to optimize the group ratios $\boldsymbol{\lambda}$, as FairBatch cannot handle $\boldsymbol{\mu}$ for the generated

data. Ideally, if the approximation error in our method is small, Dr-Fairness should obtain the same group ratios and performance as FairBatch.

Figure 3 shows that our algorithm converges to similar group ratios to those in FairBatch, although the key ideas of the two algorithms on updating the group ratio are very different. Also, the two algorithms have the similar fairness scores (0.012 equal opportunity disparity for both). These results imply that our approximations are good enough to find reasonable solutions, which is consistent with the observations in other applications (Lorraine et al., 2020; Luketina et al., 2016). We note that although FairBatch is known to have theoretical guarantees, they only apply to limited settings (e.g., binary groups and labels), so there is room to improve fairness in other settings. In the next section, we will show that Dr-Fairness achieves much higher accuracy with similar or better fairness than FairBatch on real-world datasets, as our algorithm scales to multiple groups and labels and is capable of harnessing the potential of both real and generated data.

We also verify that, in the above setting, the identity matrix approximation indeed gets almost the same group ratios with the *exact inverse Hessian* computation. See more details in Sec. A.4.

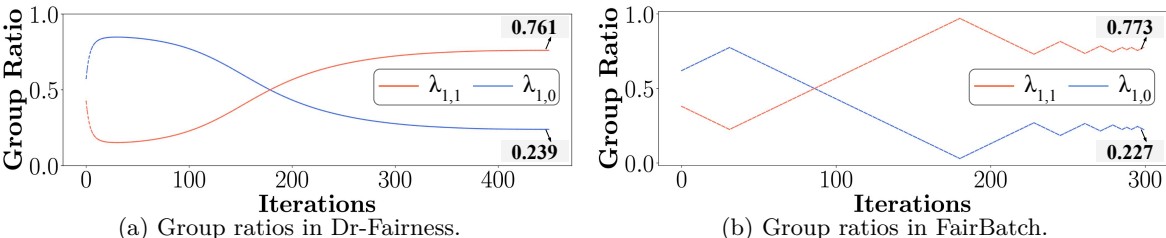

(a) Group ratios in Dr-Fairness.    (b) Group ratios in FairBatch.

Figure 3: Comparison of group ratios $\boldsymbol{\lambda}$ from Dr-Fairness and FairBatch. Both converge to similar ratios.

## 4 Experiments

We perform various experiments to evaluate our algorithm. We repeat all experiments with three random seeds and measure all performances on a separate test set – see more information in Sec. B.1. We use the Adam optimizer (Kingma & Ba, 2015).

**Datasets**  We utilize two real-world datasets: 1) CelebA (Liu et al., 2015) to compare our algorithms with baselines and perform various analyses, and 2) ImageNet People Subtree (Yang et al., 2020) to further observe the algorithm performances on a large-scale real-world scenario. Here, CelebA contains facial images, and ImageNet People Subtree provides more general form images related to people. For example, the label classes of the ImageNet People Subtree dataset include various occupations (e.g., anchor, astronaut, and tennis player), which requires the model to understand more complicated features (e.g., clothes and backgrounds) than face information to achieve a high classification performance. Note that we are using large datasets instead of the traditional smaller tabular benchmarks for fairness because our goal is to make Dr-Fairness work in large-scale real-world applications.

[*CelebA*] Contains celebrity images, where each image has 40 attributes (e.g., gender, age, and smiling). We choose group and label attributes that are less subjective and traditionally considered for fairness. The group attributes are `gender` (male and female) and `age` (young and old). The label attributes are `age`, `haircolor` (black, blond, and others), and `smiling` (smiling and not-smiling). Note that `age` can be used as either the group or label attribute. The sizes of the training/validation/test sets are 160k/20k/20k, respectively.

[*ImageNet People Subtree*] Contains 284 label classes and 3 group attributes: `gender` (male, female, and unsure), `skin color` (light, medium, and dark), and `age` (child, adult, middle, and retired). We first filter out classes that are vague, duplicates, or too small with few samples, which leaves us with 112 classes – see details in Sec. B.2. These classes contain about 111k samples, but only 10% of them have group attribute annotations. We split the group-labeled data into 40%/20%/40% for training/validation/testing, respectively.

**Data generation**  We create the generated datasets using state-of-the-art generative models that conditionally synthesize images for each $(y, z)$-class. For CelebA, we use a StyleGAN-based controllable generation

method called LACE (Nie et al., 2021). For ImageNet People Subtree, we fine-tune a diffusion model (Dhariwal & Nichol, 2021) pre-trained on ImageNet (Deng et al., 2009), and use classifier guidance (Song et al., 2020; Dhariwal & Nichol, 2021) to sample images in each $(y, z)$-class. Note that the controllable generation for ImageNet People Subtree is more challenging due to its large number of $(y, z)$-classes and labeling noises. Thus, the resulting generated data has lower quality than the generated data in CelebA. More details on data generation are in Sec. B.3.

**Baselines** We compare our algorithm with three types of baselines: 1) vanilla (non-fair) baseline, 2) fair pre-processing baselines, and 3) fair in-processing baselines.

For fair pre-processing training, we consider three baselines: *simple sampling*, *pair-augmenting (PairAug)* (Ramaswamy et al., 2021), and *pair-augmenting with our generated data (PairAug\*)*. For simple sampling, we over- and under-sample the real data to ensure an equal ratio among groups. PairAug is a fair augmentation technique that uses the generation methods to synthesize balanced images for groups to reduce the correlation between the group and label attributes. For a fair comparison, we also implement an extension of PairAug (denoted by PairAug\*), which uses the same balancing ratio in Ramaswamy et al. (2021), but uses our generated data. We note that we choose PairAug as our main baseline for the fair data augmentation strategy, as it shows state-of-the-art fairness performances among the augmentation techniques – see Sec. B.4 for a detailed comparison.

For fair in-processing training, we consider three baselines: *fairness constraint* (Zafar et al., 2017a;b), *domain independence* (Wang et al., 2020), and *FairBatch* (Roh et al., 2021). Fairness constraint adds a fairness penalty term to the loss function to reduce the unfairness. Domain independence trains separate classifiers per each group to reduce the correlation between the group and label attributes. At inference, one can ensemble the outputs of the trained classifiers to get the final predictions. FairBatch adaptively adjusts batch ratios among groups to improve fairness only using real data.

**Metrics** We focus on two accuracy metrics and three fairness metrics. [*Accuracy*] We measure the standard accuracy over all samples and the balanced accuracy that averages y-class-wise accuracies. [*Fairness*] We focus on equalized odds (EO) (Hardt et al., 2016), demographic parity (DP) (Feldman et al., 2015), and bias amplification (Zhao et al., 2017). For EO and DP, we measure the disparities (i.e., unfairness) among groups: *EO disp.* $= \max_{z \in \mathbb{Z}, y \in \mathbb{Y}} | \Pr(\hat{y}{=}y | z{=}z, y{=}y) - \Pr(\hat{y}{=}y | y{=}y) |$, and *DP disp.* $= \max_{z \in \mathbb{Z}} | \Pr(\hat{y}{=}1 | z{=}z) - \Pr(\hat{y}{=}1) |$. Together with either EO or DP, we measure bias amplification to see how much the data bias is amplified in the model: *Bias amp.* $= \max_{y \in \mathbb{Y}} \Pr(z{=}z | \hat{y}{=}y) - \Pr(z{=}z | y{=}y)$, where $z := \arg\max_{z' \in \mathbb{Z}} \Pr(z{=}z' | \hat{y}{=}y)$. Here, a good performance is indicated by high accuracy values, low EO disp. and DP disp. values, and close-to or below zero bias amp. values.

**Hyperparameters** For Dr-Fairness, we choose $k$ from a candidate set $\{0.1, 1, 10, 20\}$ to have the best fairness score while minimizing the accuracy degradation in the validation set. We set the learning rates for $\boldsymbol{\lambda}$ and $\boldsymbol{\mu}$ to 0.005. We initialize $\boldsymbol{\lambda}$ to the original $(y, z)$-ratios in the real data. We initialize $\boldsymbol{\mu}$ to 0.5 for CelebA (i.e., we start with 50% real and 50% generated data) and 0.99 for ImageNet People Subtree (i.e., 99% real and 1% generated data). We use a higher (conservative) $\boldsymbol{\mu}$ initially for ImageNet People Subtree because its generated data has lower quality than that for CelebA. For all baselines, we choose the hyperparameters from candidate sets of each baseline to show the best fairness while minimizing the accuracy degradation in the validation set.

## 4.1 CelebA Experiments

We evaluate Dr-Fairness on CelebA by comparing it with baselines (Sec. 4.1.1) and analyzing the impact of its hyperparameters (Sec. 4.1.2), components (Sec. 4.1.3), and generated data (Sec. 4.1.4).

### 4.1.1 Accuracy and Fairness

Table 2 shows the accuracy and fairness performances of different algorithms on CelebA when training w.r.t. EO (see results of training w.r.t. DP in Sec. B.5). Here we consider two scenarios: 1) binary setting of y

Table 2: Performances on the CelebA test set when training w.r.t. EO on two scenarios: binary y (age) & z (gender) and non-binary y (haircolor) & z (gender, age). We compare Dr-Fairness with three types of baselines: 1) non-fair baseline, 2) fair pre-processing baselines: Simple Sampling, PairAug, and PairAug*, and 3) fair in-processing baselines: Fair. Const., Dom. Indep., and FairBatch. We use ResNet50 (He et al., 2016) for all algorithms. Note that PairAug and Fair. Const. cannot be trivially extended to the non-binary labels, so we only show their results in the first column.

| Method | y: age and z: gender | | | | y: haircolor and z: (gender, age) | | | |
|---|---|---|---|---|---|---|---|---|
| | Acc. | Bal. Acc. | EO Disp. | Bias Amp. | Acc. | Bal. Acc. | EO Disp. | Bias Amp. |
| Non-fair | $86.4_{\pm 0.3}$ | $76.4_{\pm 0.2}$ | $0.173_{\pm 0.023}$ | $0.101_{\pm 0.022}$ | $83.8_{\pm 0.3}$ | $82.0_{\pm 0.6}$ | $0.535_{\pm 0.049}$ | $0.014_{\pm 0.003}$ |
| Simple Sampling | $86.8_{\pm 0.2}$ | $78.3_{\pm 0.5}$ | $0.132_{\pm 0.019}$ | $0.052_{\pm 0.017}$ | $83.2_{\pm 0.4}$ | $80.9_{\pm 0.3}$ | $0.421_{\pm 0.016}$ | $0.026_{\pm 0.002}$ |
| PairAug (Ramaswamy et al., 2021) | $85.6_{\pm 0.6}$ | $79.8_{\pm 0.3}$ | $0.124_{\pm 0.002}$ | $0.030_{\pm 0.003}$ | - | - | - | - |
| PairAug* (Ramaswamy et al., 2021) | $86.7_{\pm 0.2}$ | $79.3_{\pm 0.6}$ | $0.134_{\pm 0.008}$ | $0.053_{\pm 0.008}$ | $83.0_{\pm 0.2}$ | $80.1_{\pm 0.8}$ | $0.406_{\pm 0.026}$ | $0.022_{\pm 0.005}$ |
| Fair. Const. (Zafar et al., 2017b) | $86.8_{\pm 0.2}$ | $79.6_{\pm 0.5}$ | $0.106_{\pm 0.014}$ | $0.028_{\pm 0.012}$ | - | - | - | - |
| Dom. Indep. (Wang et al., 2020) | $85.0_{\pm 0.7}$ | $72.4_{\pm 2.1}$ | $0.070_{\pm 0.008}$ | $0.034_{\pm 0.021}$ | $81.7_{\pm 0.6}$ | $74.8_{\pm 1.4}$ | $0.340_{\pm 0.041}$ | $0.026_{\pm 0.013}$ |
| FairBatch (Roh et al., 2021) | $84.7_{\pm 0.3}$ | $72.5_{\pm 1.8}$ | $0.023_{\pm 0.014}$ | $\mathbf{-0.043_{\pm 0.007}}$ | $79.4_{\pm 1.5}$ | $68.2_{\pm 4.7}$ | $0.096_{\pm 0.033}$ | $0.033_{\pm 0.003}$ |
| **Dr-Fairness** | $\mathbf{87.7_{\pm 0.3}}$ | $\mathbf{81.0_{\pm 0.2}}$ | $\mathbf{0.020_{\pm 0.010}}$ | $-0.026_{\pm 0.002}$ | $\mathbf{85.0_{\pm 0.1}}$ | $\mathbf{84.4_{\pm 0.3}}$ | $\mathbf{0.079_{\pm 0.023}}$ | $\mathbf{0.012_{\pm 0.002}}$ |

(age) & z (gender) and 2) non-binary setting of y (haircolor) & z (gender, age). In Sec. B.6, we show similar results for the experiments on other group and label combinations. Also, in Sec. B.7, we visually demonstrate the accuracy-fairness tradeoffs of the algorithms.

The fair pre-processing baselines (in rows 2–4) improve the fairness performances compared to the original non-fair baseline, but still perform worse (i.e., higher EO disp. and higher bias amp.) than the fair in-processing baselines and Dr-Fairness. Thus, simply equalizing the data ratio among groups may not be enough to achieve high group fairness. Note that it is not straightforward to get the generated data from the original PairAug work in the non-binary label setting, so we are not able to report the numbers (e.g., the right columns of Table 2). But we expect that the results would be similar to PairAug*, as observed in the binary setting. Additionally, FairnessGAN (Sattigeri et al., 2019) is another previous method that aims to generate fair images, but this method has been reported to show worse fairness and accuracy performances than PairAug – see Sec. B.4 for a detailed comparison.

The fair in-processing baselines (in rows 5–7) improve fairness (esp. EO), but tend to sacrifice accuracy because they only utilize real data. Here, in the baselines with higher fairness, the decrease in accuracy becomes more significant. For example, FairBatch adaptively adjusts the group ratio on real data to improve fairness, but we observe that some small-sized groups end up being oversampled, which is detrimental to the accuracy performance on the test set.

In comparison, Dr-Fairness achieves high fairness performances while even improving accuracies by adaptively finding optimal data ratios among groups and between real and generated data. There are **two takeaways**: 1) we can find a better group ratio than the 1:1 ratio for fairness, and 2) an optimal combination of real and generated data can mitigate the accuracy degradation of fair training.

**Remark 5.** *We explain when Dr-Fairness can improve both fairness and accuracy compared to other fairness baselines. We believe this phenomenon is related to the optimal accuracy-fairness tradeoff, which is known to be determined by the data distribution (Menon & Williamson, 2018; Roh et al., 2023). When the performance of a fair algorithm lies on the optimal accuracy-fairness tradeoff, any other algorithm can only achieve either better fairness or better accuracy, but cannot improve both. However, when the fairness algorithms do not achieve the optimal accuracy-fairness tradeoff in the given data, there is an opportunity to improve the model's performances toward the optimal tradeoff. For example, we observe that Dr-Fairness improves both accuracy and fairness compared to other baselines in CelebA, implying that the baselines do not achieve the optimal accuracy-fairness tradeoff in the first place – see the tradeoff curve comparisons in Sec. B.7.*

### 4.1.2 Hyperparameter Analysis

We now evaluate Dr-Fairness by varying its main hyperparameter $k$ used in the bilevel optimization. A larger $k$ puts more weight on the accuracy loss than the fairness loss. Figures 4a and 4b show the accuracy and

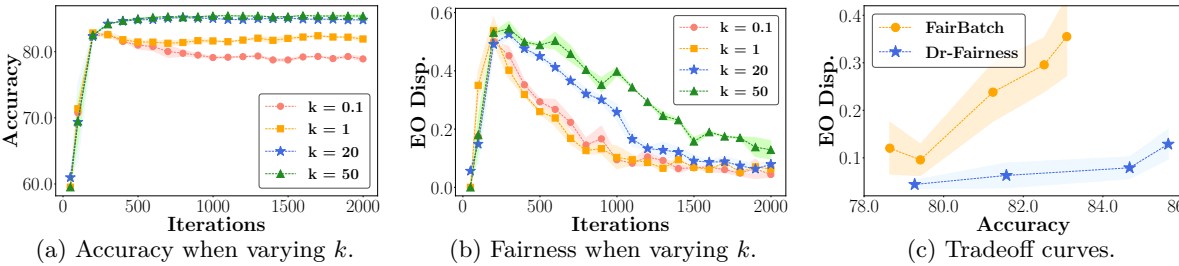

(a) Accuracy when varying $k$.     (b) Fairness when varying $k$.     (c) Tradeoff curves.

Figure 4: The performances of Dr-Fairness by varying the hyperparameter $k$ when (y, z) = (haircolor, (gender, age)). The first two graphs show the performance changes during the training, and the last graph shows the accuracy-fairness tradeoff curves. Compared to FairBatch, Dr-Fairness shows a better tradeoff.

fairness of Dr-Fairness during the training with the different $k$ values. As expected, increasing $k$ (say $k = 50$) results in higher accuracy and lower fairness. By varying $k$, we can also compare the accuracy-fairness tradeoff curves of Dr-Fairness and FairBatch in Figure 4c. Dr-Fairness shows a better tradeoff, which is consistent with the results in Sec. 4.1.1.

### 4.1.3 Ablation Study

We perform an ablation study on our framework to evaluate the impact of each component in the optimization on fairness and accuracy. For fairness, we conduct two ablations: F1) remove both the fairness loss in the outer objective and $\lambda_{y,z}$ in the inner objective, and F2) only remove the fairness loss in the outer objective. For accuracy, we consider three ablations: A1) remove the accuracy loss in the outer objective; and $\mu_{y,z}$ and the generated data loss in the inner objective, A2)

Table 3: Ablation study on CelebA, where we consider the setting of non-binary y (haircolor) and binary z (gender). We mark noticeable *performance degradations* with underlines.

| Method | Acc. | Bal. Acc. | EO Disp. |
|---|---|---|---|
| F1: w/o fair. loss and $\lambda_{y,z}$ | $85.7_{\pm 0.3}$ | $83.9_{\pm 1.0}$ | $\underline{0.432_{\pm 0.063}}$ |
| F2: w/o fair. loss | $85.1_{\pm 0.3}$ | $84.6_{\pm 0.3}$ | $\underline{0.261_{\pm 0.017}}$ |
| A1: w/o acc. loss, $\mu_{y,z}$, and gen. data | $\underline{79.3_{\pm 1.2}}$ | $\underline{66.7_{\pm 3.8}}$ | $0.090_{\pm 0.058}$ |
| A2: w/o $\mu_{y,z}$, and gen. data | $\underline{80.0_{\pm 1.2}}$ | $\underline{70.8_{\pm 3.1}}$ | $0.031_{\pm 0.012}$ |
| A3: w/o acc. loss | $\underline{71.5_{\pm 0.9}}$ | $82.1_{\pm 0.8}$ | $0.046_{\pm 0.029}$ |
| **Dr-Fairness** | $84.5_{\pm 0.2}$ | $82.3_{\pm 0.7}$ | $0.029_{\pm 0.009}$ |

remove $\mu_{y,z}$ and the generated data loss in the inner objective, and A3) only remove the accuracy loss in the outer objective. We note that A2 also represents how Dr-Fairness works only with real data when we cannot utilize generated data. Through this sequence of ablations, we observe that each part of our algorithm gradually improves the fairness and accuracy performances.

In Table 3, the fairness ablations (in rows 1–2) show worse fairness as the fairness loss and $\lambda_{y,z}$ are discarded, and the accuracy ablations (in rows 3–5) demonstrate lower accuracy and balanced accuracy as some of the accuracy loss, generated data, and $\mu_{y,z}$ are removed. We thus conclude that all components in our bilevel optimization contribute to the overall performances.

### 4.1.4 Generated Data of Different Qualities

We analyze the robustness of our framework against the generated data quality as shown in Table 4. We vary the quality of generated data by adding random Gaussian noise to the original images. Interestingly, when the generated data quality decreases (i.e., adding more noise to the images), Dr-Fairness automatically reduces the usage of the generated data, as shown in the second column of Table 4. With this automatic adjustment, Dr-Fairness shows robust performances in the last three columns. When the generated data is fully replaced with Gaussian noise (i.e., severe

Table 4: Analysis of the behavior of Dr-Fairness when the quality of generated data changes. We add different amounts of Gaussian noise to the original (clean) generated data. In the severe case, we fully replace the generated data with the noise. We set y to haircolor (non-binary) and z to gender (binary).

| Noise level in gen. data | Final gen. data ratio found in Dr-Fairness | Acc. | Bal. Acc. | EO Disp. |
|---|---|---|---|---|
| Clean | $0.224_{\pm 0.009}$ | $84.5_{\pm 0.2}$ | $82.3_{\pm 0.7}$ | $0.029_{\pm 0.009}$ |
| Light noise | $0.210_{\pm 0.018}$ | $84.4_{\pm 0.2}$ | $81.6_{\pm 1.0}$ | $0.033_{\pm 0.003}$ |
| Mid noise | $0.192_{\pm 0.012}$ | $84.5_{\pm 0.1}$ | $81.9_{\pm 0.3}$ | $0.039_{\pm 0.022}$ |
| Severe noise | $0.114_{\pm 0.041}$ | $83.1_{\pm 1.8}$ | $78.3_{\pm 0.8}$ | $0.108_{\pm 0.038}$ |
| Non-fair baseline | – | $83.8_{\pm 0.3}$ | $82.0_{\pm 0.6}$ | $0.399_{\pm 0.019}$ |

noise), the accuracy and fairness performances become worse than the clean setting as expected, but the

fairness score is still much better than the non-fair baseline by reasonably sacrificing the accuracy. These results show that Dr-Fairness is effective even with low-quality generated data.

## 4.2 ImageNet People Subtree Experiments

We finally perform experiments on ImageNet People Subtree, which represents a large-scale real-world scenario with 112 label classes and 3 non-binary group attributes, gender, skin color, and age. As only 10% of the data has group annotations, following Zhao et al. (2021), we first pre-train a non-fair model on the entire training set with $y$ labels and then fine-tune the pre-trained model to improve fairness on the small set with group labels.

Tables 5 and 6 show the performances of the algorithms on four group scenarios: gender, skin color, age, and all combinations of them. The overall results are consistent with the CelebA experiments, where we can see Dr-Fairness outperforms the baselines in accuracy, fairness, or both. Specifically, our algorithm shows the best or second-best performance on EO disparity and bias amplification in almost all group settings while obtaining better classification accuracies compared to the baselines with similar fairness scores. For example, we obtain classification accuracies better than FairBatch, with an absolute improvement of 5–9%, while achieving similar fairness scores.

As ImageNet People Subtree shows a more complicated real-world scenario than CelebA, we have two additional observations. First, when we train the baselines w.r.t. EO, the bias amplification metric occasionally gets worse compared to the original model (e.g., Dom. Indep. on gender and FairBatch on skin color). This result shows that improving EO, which aims to minimize the label-specific accuracy gap between groups, does not necessarily lead to reducing the bias in the model compared to the data. In addition, as domain independence trains separate classifiers per each group, we suspect that the final model may have undesirable results (e.g., worse bias amp.) if some of the classifiers fail. Second, this dataset contains a large number of $(y, z)$-classes where many of them are extremely small-sized. Here the controllable data generation becomes challenging where the generated labels may be noisy, which negatively affects the fair training as well. Nonetheless, Dr-Fairness still shows a clear improvement in fairness compared to the baselines, and we believe more data with clean labels could further improve its performance.

Table 5: Performances on the ImageNet People Subtree test set when training w.r.t. equalized odds (EO) for either gender or skin color. We mark the best and second best performances among the fairness algorithms with bold and underline, respectively. Note that PairAug and Fair. Const. used in Table 2 cannot be trivially extended to the non-binary labels, so we do not use them on ImageNet People Subtree. Other settings are identical to Table 2.

| Method | z: gender | | | | z: skin color | | | |
|---|---|---|---|---|---|---|---|---|
| | Acc. | Bal. Acc. | EO Disp. | Bias Amp. | Acc. | Bal. Acc. | EO Disp. | Bias Amp. |
| Non-fair | $61.4_{\pm0.8}$ | $61.8_{\pm0.8}$ | $0.871_{\pm0.013}$ | $0.256_{\pm0.031}$ | $61.4_{\pm0.8}$ | $61.8_{\pm0.8}$ | $0.874_{\pm0.024}$ | $0.239_{\pm0.062}$ |
| Simple Sampling | $54.8_{\pm0.8}$ | $54.9_{\pm0.9}$ | $0.834_{\pm0.004}$ | $0.282_{\pm0.038}$ | $55.5_{\pm1.1}$ | $55.7_{\pm1.0}$ | $0.849_{\pm0.000}$ | $0.219_{\pm0.035}$ |
| PairAug* (Ramaswamy et al., 2021) | $54.6_{\pm0.5}$ | $54.9_{\pm0.5}$ | $0.821_{\pm0.030}$ | $0.241_{\pm0.034}$ | $58.1_{\pm0.1}$ | $58.3_{\pm0.1}$ | $\underline{0.830}_{\pm0.000}$ | $0.225_{\pm0.010}$ |
| Dom. Indep. (Wang et al., 2020) | $\mathbf{59.9}_{\pm0.1}$ | $\mathbf{60.1}_{\pm0.1}$ | $0.857_{\pm0.016}$ | $0.381_{\pm0.033}$ | $\underline{60.2}_{\pm0.2}$ | $\underline{60.5}_{\pm0.3}$ | $0.874_{\pm0.009}$ | $0.204_{\pm0.009}$ |
| FairBatch (Roh et al., 2021) | $52.9_{\pm3.1}$ | $53.1_{\pm3.2}$ | $\mathbf{0.816}_{\pm\mathbf{0.076}}$ | $\mathbf{0.191}_{\pm\mathbf{0.011}}$ | $51.9_{\pm0.5}$ | $52.1_{\pm0.5}$ | $\mathbf{0.811}_{\pm\mathbf{0.019}}$ | $0.346_{\pm0.021}$ |
| **Dr-Fairness** | $\underline{58.2}_{\pm0.3}$ | $\underline{58.3}_{\pm0.2}$ | $\underline{0.817}_{\pm0.012}$ | $\underline{0.220}_{\pm0.027}$ | $\mathbf{60.8}_{\pm\mathbf{1.4}}$ | $\mathbf{61.2}_{\pm\mathbf{1.3}}$ | $0.845_{\pm0.001}$ | $\mathbf{0.137}_{\pm\mathbf{0.009}}$ |

The above experiments use ResNet50 (He et al., 2016) as the model backbone. In addition, we also conduct experiments using ViT (Dosovitskiy et al., 2021) and observe similar results. Specifically, we use DeiT-S (Touvron et al., 2021), a variant of ViT (Dosovitskiy et al., 2021), on the ImageNet People Subtree dataset w.r.t. the age group attribute. Table 7 shows the accuracy and fairness performances of Dr-Fairness and the representative baselines, where we observe similar results to those in Tables 5 and 6. This experiment shows that Dr-Fairness is applicable to various network architectures, including ViT.

Table 6: Performances on the ImageNet People Subtree test set when training w.r.t. equalized odds (EO) for either age or all combinations of groups. Other settings are identical to Table 2.

| | z: age | | | | z: (gender, skin color, age) | | | |
|---|---|---|---|---|---|---|---|---|
| Method | Acc. | Bal. Acc. | EO Disp. | Bias Amp. | Acc. | Bal. Acc. | EO Disp. | Bias Amp. |
| Non-fair | $61.4_{\pm0.8}$ | $61.8_{\pm0.8}$ | $0.849_{\pm0.029}$ | $0.191_{\pm0.039}$ | $61.4_{\pm0.8}$ | $61.8_{\pm0.8}$ | $0.918_{\pm0.016}$ | $0.270_{\pm0.107}$ |
| Simple Sampling | $53.4_{\pm0.9}$ | $53.6_{\pm0.9}$ | $0.822_{\pm0.044}$ | $0.201_{\pm0.077}$ | $54.6_{\pm0.9}$ | $54.8_{\pm0.8}$ | $0.885_{\pm0.041}$ | $0.181_{\pm0.002}$ |
| PairAug* (Ramaswamy et al., 2021) | $55.6_{\pm1.2}$ | $55.9_{\pm1.3}$ | $0.820_{\pm0.046}$ | $\underline{0.168_{\pm0.022}}$ | $57.6_{\pm0.4}$ | $57.9_{\pm0.5}$ | $0.871_{\pm0.005}$ | $\underline{0.153_{\pm0.036}}$ |
| Dom. Indep. (Wang et al., 2020) | $\mathbf{59.9_{\pm0.2}}$ | $\mathbf{60.1_{\pm0.3}}$ | $0.838_{\pm0.019}$ | $0.220_{\pm0.006}$ | $50.0_{\pm0.9}$ | $50.1_{\pm0.9}$ | $0.897_{\pm0.015}$ | $0.254_{\pm0.020}$ |
| FairBatch (Roh et al., 2021) | $51.3_{\pm1.2}$ | $51.5_{\pm1.2}$ | $\underline{0.798_{\pm0.042}}$ | $0.198_{\pm0.003}$ | $50.1_{\pm5.3}$ | $50.3_{\pm5.3}$ | $\mathbf{0.853_{\pm0.013}}$ | $0.185_{\pm0.021}$ |
| **Dr-Fairness** | $\underline{59.7_{\pm0.1}}$ | $\underline{59.9_{\pm0.1}}$ | $\mathbf{0.784_{\pm0.011}}$ | $\mathbf{0.149_{\pm0.012}}$ | $\mathbf{58.7_{\pm0.5}}$ | $\mathbf{59.1_{\pm0.5}}$ | $\underline{0.866_{\pm0.012}}$ | $\mathbf{0.146_{\pm0.006}}$ |

Table 7: Performances on the ImageNet People Subtree test set w.r.t. equalized odds (EO). We use DeiT-S (Touvron et al., 2021), a variant of ViT (Dosovitskiy et al., 2021), for all algorithms. Other settings are identical to Table 2.

| | z: age | | | |
|---|---|---|---|---|
| Method | Acc. | Bal. Acc. | EO Disp. | Bias Amp. |
| Non-fair | $64.8_{\pm0.0}$ | $65.2_{\pm0.1}$ | $0.890_{\pm0.027}$ | $0.198_{\pm0.030}$ |
| Simple Sampling | $55.3_{\pm0.7}$ | $55.7_{\pm0.7}$ | $\underline{0.807_{\pm0.011}}$ | $0.261_{\pm0.081}$ |
| PairAug* (Ramaswamy et al., 2021) | $\underline{59.5_{\pm0.1}}$ | $\underline{59.9_{\pm0.1}}$ | $0.824_{\pm0.006}$ | $0.237_{\pm0.083}$ |
| Dom. Indep. (Wang et al., 2020) | $51.8_{\pm1.6}$ | $52.0_{\pm1.5}$ | $0.817_{\pm0.005}$ | $0.634_{\pm0.246}$ |
| FairBatch (Roh et al., 2021) | $58.2_{\pm0.6}$ | $58.6_{\pm0.5}$ | $0.817_{\pm0.001}$ | $\mathbf{0.162_{\pm0.016}}$ |
| **Dr-Fairness** | $\mathbf{64.4_{\pm0.6}}$ | $\mathbf{64.8_{\pm0.6}}$ | $\mathbf{0.774_{\pm0.001}}$ | $\underline{0.176_{\pm0.001}}$ |

## 5 Conclusion

We proposed a novel adaptive sampling approach called Dr-Fairness that utilizes both real and generated data for fairness. To perform adaptive sampling systematically, we first formulated a bilevel optimization, where the goal is to find the optimal data ratios among sensitive groups and between real and generated data to achieve high group fairness while minimizing accuracy degradation. To solve the bilevel optimization problem, we then designed an efficient approximate algorithm based on the implicit function theorem and identity-matrix approximation. Extensive experiments on the CelebA and ImageNet People Subtree datasets showed that Dr-Fairness achieves state-of-the-art fairness and accuracy performances. In addition, analyzing the performance of Dr-Fairness in other types of datasets, including medical images, will be an interesting future direction. We believe Dr-Fairness opens up new opportunities for effectively using generated data in large-scale real-world scenarios.

**Broader Impact Statement**

We believe our work can positively impact society by reducing discrimination in AI applications. In particular, our framework shows that generated data can compensate for unfairness issues in real data (e.g., size bias and lack of diversity) to help obtain better accuracy and fairness results that would not have been possible otherwise. As a result, real-world applications have a better chance of ensuring fairness without sacrificing accuracy unnecessarily.

We do note that choosing an appropriate fairness metric for each application is essential, as a poor choice may lead to unintended discrimination. Thus, one needs to carefully choose the target fairness metrics based on the social context in each application. Also, in terms of privacy, we did not involve human subjects or use any direct personal identifiers in the experiments, except for the human images in the publicly available benchmark datasets.

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

# A    Appendix – Optimization and Algorithm

## A.1    Bilevel Optimization for Demographic Parity

Continuing from Sec. 3.1, we formulate our bilevel optimization w.r.t. demographic parity (DP) as follows:

$$\min_{\boldsymbol{\lambda},\boldsymbol{\mu}} \max_{y\in\mathbb{Y},z_1,z_2\in\mathbb{Z}}\{|\tfrac{m_{y,z_1}}{m_{\star,z_1}}L_{y,z_1}^{\text{real}}(\boldsymbol{w}(\boldsymbol{\lambda},\boldsymbol{\mu})) - \tfrac{m_{y,z_2}}{m_{\star,z_2}}L_{y,z_2}^{\text{real}}(\boldsymbol{w}(\boldsymbol{\lambda},\boldsymbol{\mu}))|\} + k\sum_{y\in\mathbb{Y},z\in\mathbb{Z}}\tfrac{m_{y,z}}{m}L_{y,z}^{\text{real}}(\boldsymbol{w}(\boldsymbol{\lambda},\boldsymbol{\mu})),$$

$$\boldsymbol{w}(\boldsymbol{\lambda},\boldsymbol{\mu}) = \arg\min_{\boldsymbol{w}} \sum_{y\in\mathbb{Y},z\in\mathbb{Z}}\frac{m_{y,\star}}{m}\lambda_{y,z}\{\mu_{y,z}L_{y,z}^{\text{real}}(\boldsymbol{w}) + (1-\mu_{y,z})L_{y,z}^{\text{gen}}(\boldsymbol{w})\},$$

$$\text{s.t.}\quad \boldsymbol{\lambda}\in[0,1], \boldsymbol{\mu}\in[0,1], \ \textstyle\sum_{z\in\mathbb{Z}}\lambda_{y,z} = 1, \forall y\in\mathbb{Y},$$

where $\mathbb{Y} = \{0,1\}$. Note that DP is designed for binary classification. For designing the fairness loss, we are inspired by Roh et al. (2021), which gives a hint on formulating DP loss in bilevel optimization. Intuitively, the fractions in the fairness loss make the model reduces the disparity of each prediction ratio across groups, without considering the sizes of true label classes. This strategy can be a sufficient condition for DP, as the goal of DP is to achieve the same positive prediction ratio among groups – see more details in Roh et al. (2021).

## A.2    Training Overhead for Computing Gradients

Continuing from Sec. 3.2, we empirically analyze the training overhead when computing gradients in Eq. 3. Figure 5 shows the relationship between update frequency and training time on the synthetic dataset – see the details on the dataset in Sec. A.3. We compare Dr-Fairness with vanilla (non-fair) training, which shows a basic training cost, and we use the same total number of training iterations for both algorithms. As a result, we can reduce the computational cost of Dr-Fairness by updating $\boldsymbol{\lambda}$ and $\boldsymbol{\mu}$ less frequently (i.e., using a longer update period). For example, if we update $\boldsymbol{\lambda}$ and $\boldsymbol{\mu}$ at every two iterations, the training time reduces by about half compared to when updating them at every single iteration.

In addition, Figure 6 shows that we can reduce the training overhead without sacrificing the overall accuracy and fairness performances much. Here, when updating the ratios less frequently, we can achieve comparable performances using the same total iterations by increasing the learning rates used in the optimizers for $\boldsymbol{\lambda}$ and $\boldsymbol{\mu}$, but this may also lead to a larger standard deviation.

Finally, we observe consistent results in the CelebA dataset, where we consider the setting of binary y (age) and z (gender). Specifically, we can reduce the training time by about half by updating $\boldsymbol{\lambda}$ and $\boldsymbol{\mu}$ at every two iterations, while the overall accuracy and fairness performances remain similar to those when updating the data ratios at every single iteration, as shown in Table 8. Similar to the above synthetic data experiment, we increase the learning rates used in the optimizers for $\boldsymbol{\lambda}$ and $\boldsymbol{\mu}$ when updating them less frequently.

Table 8: Performances on the CelebA test set when training w.r.t. EO on binary y (age) and z (gender). We compare the performances when varying the update period of $\boldsymbol{\lambda}$ and $\boldsymbol{\mu}$ in Dr-Fairness.

| Method | Update Period of $\boldsymbol{\lambda}$ & $\boldsymbol{\mu}$ | Acc. | Bal. Acc. | EO Disp. | Bias Amp. |
|---|---|---|---|---|---|
| Non-fair | – | $86.4_{\pm 0.3}$ | $76.4_{\pm 0.2}$ | $0.173_{\pm 0.023}$ | $0.101_{\pm 0.022}$ |
| **Dr-Fairness** | 1 iteration | $87.7_{\pm 0.3}$ | $81.0_{\pm 0.2}$ | $0.020_{\pm 0.010}$ | $-0.026_{\pm 0.002}$ |
| **Dr-Fairness** | 2 iterations | $87.6_{\pm 0.1}$ | $81.1_{\pm 0.2}$ | $0.025_{\pm 0.003}$ | $-0.024_{\pm 0.001}$ |

## A.3    Setting for the Validity Check

Continuing from Sec. 3.2, we describe the synthetic binary setting in Roh et al. (2021), which is used to empirically verify how close the solutions from our approximation strategy are to the optimal ones.

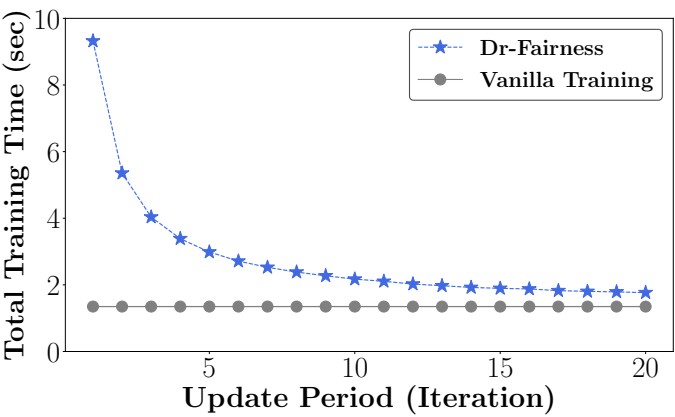

Figure 5: Update-period vs training-time curves on the synthetic dataset. We compare Dr-Fairness with vanilla (non-fair) training, which shows a basic training cost.

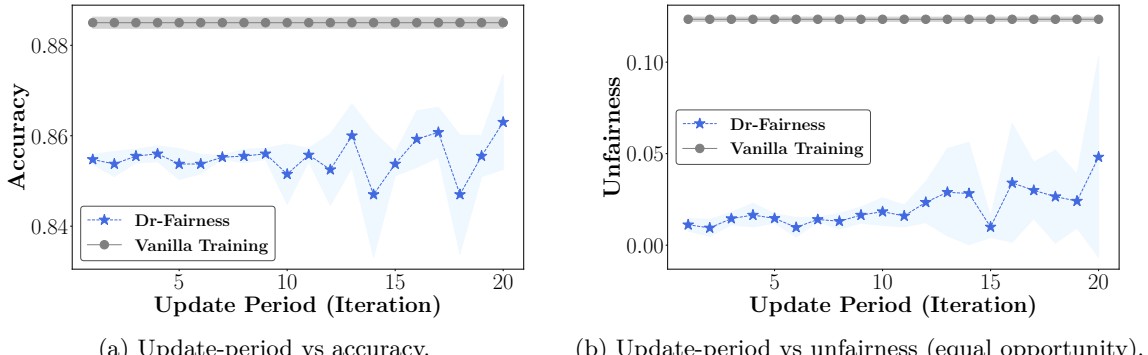

(a) Update-period vs accuracy.  (b) Update-period vs unfairness (equal opportunity).

Figure 6: Performance comparisons when varying the update period in Dr-Fairness on the synthetic dataset. We compare Dr-Fairness with vanilla (non-fair) training.

For generating the synthetic dataset, we use a method in Zafar et al. (2017a), which produces two input attributes $(x_1, x_2)$, one binary label attribute y, and one binary group attribute z. We draw each sample $(x_1, x_2, y)$ from Gaussian distributions and make z follow a biased distribution.

In detail, we generate each sample $(x_1, x_2, y)$ from two Gaussian distributions: $(x_1, x_2)|y = 0 \sim \mathcal{N}([-2; -2], [10, 1; 1, 3])$ and $(x_1, x_2)|y = 1 \sim \mathcal{N}([2; 2], [5, 1; 1, 5])$. Then, we make z follow a biased distribution: $\Pr(z = 1) = \Pr((x_1', x_2')|y = 1)/[\Pr((x_1', x_2')|y = 0) + \Pr((x_1', x_2')|y = 1)]$ where $(x_1', x_2') = (x_1 \cos(\pi/4) - x_2 \sin(\pi/4), x_1 \sin(\pi/4) + x_2 \cos(\pi/4))$. This synthetic dataset contains training, validation, and test sets with 2k, 1k, and 1k samples, respectively.

In this experiment, we use logistic regression models for all algorithms, as in Roh et al. (2021).

### A.4  Comparison with Exact Inverse Hessian Computation

Continuing from Sec. 3.2, we compare our identity matrix-based approximation results with the exact inverse Hessian computation results. We use the same setting described in Sec. A.3, where it is tractable to compute the exact inverse Hessian.

Figure 7 shows the group ratios of Dr-Fairness (with the identity matrix approximation) and Dr-Fairness with the exact inverse Hessian computation. We can see that Dr-Fairness, which uses the identity matrix approximation to estimate the inverse Hessian in Eq. 2, converges to similar group ratios to those in computing the exact inverse Hessian. It implies that our solution is close to the exact solution despite the method's simplicity. Another observation is that the data ratios in Figure 7b converge within fewer iterations than

those in Figure 7a. We note that although the number of required iterations is fewer when computing the exact inverse Hessian, the training time is much slower if the number of parameters increases. Thus, the approximation used in Dr-Fairness can be a reasonable solution to estimate the inverse Hessian, which is usually intractable for large models and data.

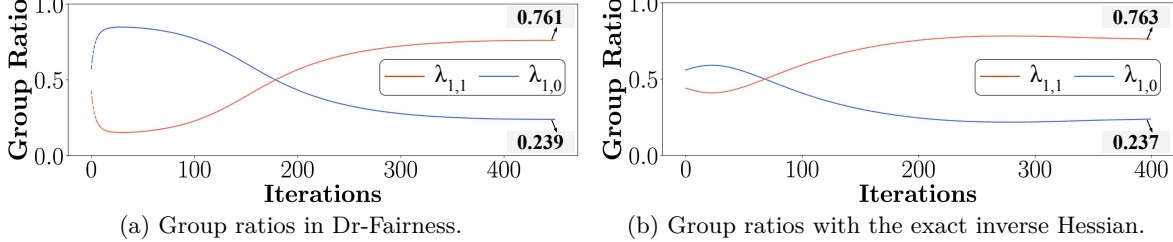

(a) Group ratios in Dr-Fairness.       (b) Group ratios with the exact inverse Hessian.

Figure 7: Comparison of group ratios $\boldsymbol{\lambda}$ from Dr-Fairness (with the identity matrix approximation) and Dr-Fairness with the exact inverse Hessian computation. Both converge to similar ratios.

## A.5   Comparison with Other Problem Formulation Methods

Continuing from Sec. 3.1, we discuss the advantages of using bilevel optimization compared to other problem formulation methods, especially using distributionally robust optimization (DRO) (Sinha et al., 2017).

DRO is one of the prominent problem formulation methods in machine learning, which can solve a target objective in a min-max formulation, but we believe that our bilevel formulation is more suitable to handle both real and generated data while improving group fairness. In our scenario, real data and generated data play very different roles in the bilevel objectives, and such roles are difficult to capture via a DRO formulation.

- In detail, at the test-time evaluation, we only care about the fairness and accuracy losses (in our outer objective) on the real data distribution. Therefore, the empirical risk for generated data ($L^{\text{gen}}$) only appears in the inner objective for model parameters update.

- Directly applying DRO to empirical risks of real and generated data does not lead to the same effect as our bilevel objective because this ignores the EO-based loss in our outer objective, and more importantly, it is unclear what is the benefit of optimizing the real/generated data sampling ratio to maximize the empirical risks. If we consider an example where we have a high loss on generated data and a low loss on real data, then DRO should increase the sampling ratio for the generated data to increase the overall loss. However, the high loss on generated data could be the result of the low quality of generated data, and increasing its sampling ratio might instead hurt the performance and fairness. On the other hand, as shown in Sec. 4.1.4, Dr-Fairness would decrease the ratio of generated data when its quality is low.

# B   Appendix – Experiments

## B.1   Experimental Settings

Continuing from Sec. 4, we provide detailed information on the experimental settings. We use PyTorch for all experiments and utilize the pre-trained ResNet50 (He et al., 2016) provided by PyTorch library (i.e., torchvision (Marcel & Rodriguez, 2010)). We change the last fully-connected layer of each model with the number of corresponding label classes. When training, we update all model parameters in the pre-trained model. The batch size of all experiments is 128. We set the learning rate for updating model parameters to 0.0001. For the data ratio ($\boldsymbol{\lambda}$ and $\boldsymbol{\mu}$) updates in Dr-Fairness, we use the Adam optimizer and set the learning rate for the ratio update to 0.005 in all experiments. When calculating the gradients w.r.t. model parameters or data ratios in Eq. 3, we use the autograd functionality in PyTorch. We apply the exponential

moving average when updating the model parameters in Dr-Fairness. To prevent the overfitting, we use the validation set when measuring the fairness and accuracy losses in the outer objective of our bilevel optimization. Similarly, we use the validation set in other baselines if they require the computation of additional (fairness) losses in the algorithm.

## B.2  Filtering Label Classes in ImageNet People Subtree

Continuing from Sec. 4, we explain how we filter the label classes in the ImageNet People Subtree dataset. Initially, the dataset contains 284 label classes. Among them, we filter out classes that are vague, duplicates, or too small with few samples. First, we filter vague classes like "ex-president" and "junior", which are hard to classify even for human annotators. To decide whether each class is vague or not, we perform internal crowdsourcing. For each class, we gather 3 expert decisions and do a majority vote. Also, we remove classes that are conceptually duplicates of others. Finally, we set the allowed minimum sample size to 50 and ignore the classes with fewer than 50 samples. As a result, 112 classes are used in our experiments.

## B.3  Data Generation

Continuing from Sec. 4, we explain the details on data generation.

For experiments on CelebA, we use LACE (Nie et al., 2021) to generate data. LACE is a controllable generation method that uses an energy-based model (EBM) in the latent space of a pre-trained generative model such as StyleGAN2 (Karras et al., 2020). We consider StyleGAN2 pre-trained on the CelebA-HQ dataset as our base generative model. In LACE, we first need to train the latent classifiers in the $w$-space of StyleGAN2, each of which corresponds to an energy function for an individual attribute in the EBM formulation (see Eq. (4) in (Nie et al., 2021)). Since we mainly focus on five attributes (i.e., `age`, `gender`, `smile`, `glasses`, and `haircolor`) in the CelebA experiments, we end up with five latent classifiers. Next, for each combination of attribute values (e.g., `age`='young', `gender`='female', `smile`='true', `glasses`='true', and `haircolor`='black'), we use the ordinal differential equation (ODE) sampler in the latent space to sample the corresponding images. We repeat the above sampling process until we cover all the combinations of attribute values.

For experiments on ImageNet People Subtree, we use a guided diffusion model (Dhariwal & Nichol, 2021) with classifier guidance (Song et al., 2020; Dhariwal & Nichol, 2021) to generate data. Since there exist no ADM checkpoints pre-trained on ImageNet People Subtree, we first fine-tune the ImageNet-pretrained ADM on ImageNet People Subtree, where the ADM model that we use conditions on the 112 labels. For classifier guidance, we also need to first train three time-dependent attribute classifiers, each corresponding to a demographic attribute (i.e., `gender`, `skin color`, and `age`), on noisy images produced by the diffusion process. In particular, we fine-tune the noisy image classifier (also pre-trained on ImageNet) with three new prediction heads on 10% of the annotated data. Next, for each combination of label and attribute values, we pass the label value as the input of the conditional ADM and use the classifier guidance (i.e., the guidance from the fine-tune noisy image classifier in Eq. (10) of Dhariwal & Nichol (2021)) with the scale $s = 15$. Similarly, we repeat the above sampling process until we cover all the combinations of label and attribute values.

Here we describe the number of generated samples in each dataset. In CelebA, we consider 5 attributes for the controllable generation: `gender` (male and female), `age` (young and old), `smile` (true and false), `glasses` (true and false), and `haircolor` (black, blond, and others). Thus, these 5 attributes yield 48 class combinations (i.e., $2^4 \times 3$). We generate a total of 96k samples, where there are 2k samples for each attribute combination (e.g., 2k samples for (`age`='young', `gender`='female', `smile`='true', `glasses`='true', and `haircolor`='black')). In ImageNet People Subtree, there are 112 label classes and 3 group attributes: `gender` (male, female, and unsure), `skin color` (light, medium, and dark), and `age` (child, adult, middle, and retired). Thus, we have 4,032 combinations (i.e., $112 \times 3 \times 3 \times 4$) for the controllable generation. We generate 32 samples for each combination, which results in about 129k samples in total.

Figures 8 and 9 show examples of the generated images. We note that the controllable generation for ImageNet People Subtree is more challenging due to its large number of $(y, z)$-classes and labeling noises. Thus, the resulting generated data is noisier than the generated data in CelebA.

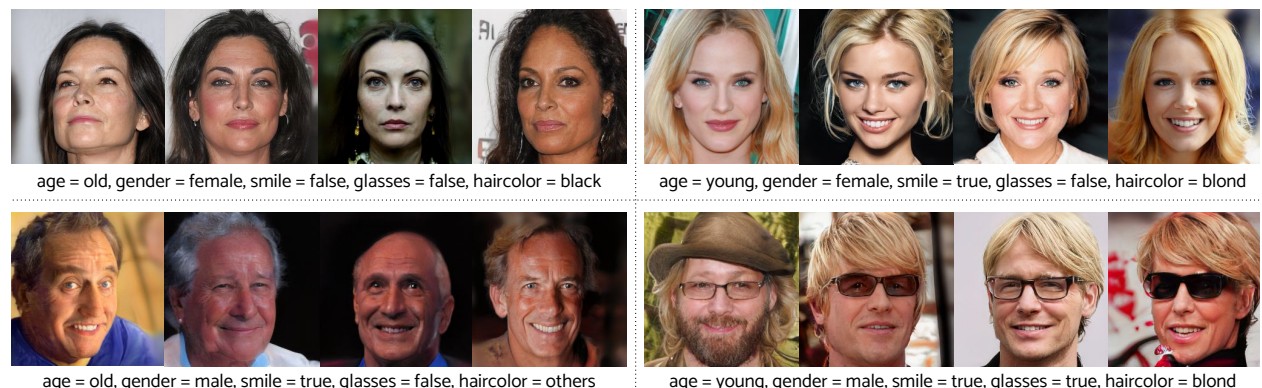

Figure 8: Examples of the generated images on CelebA.

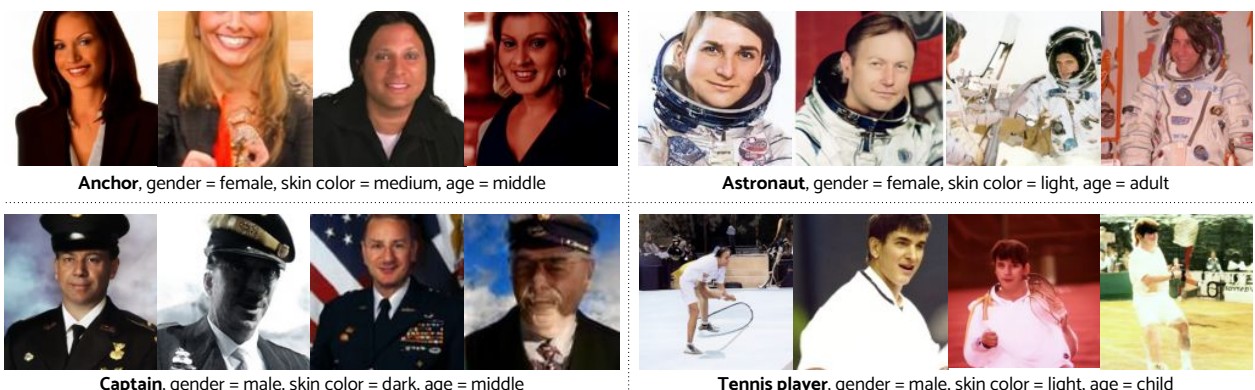

Figure 9: Examples of the generated images on ImageNet People Subtree.

## B.4 Comparison between FairnessGAN and PairAug

Continuing from Sec. 4.1.1, we compare FairnessGAN (Sattigeri et al., 2019) and PairAug (Ramaswamy et al., 2021). Table 9 shows the accuracy and fairness performances of the two algorithms on CelebA, where they consider the setting of binary label attribute y (attractive) and binary group attribute z (gender). We show the numbers that are reported in Ramaswamy et al. (2021). As a result, PairAug shows better accuracy and equalized odds performances compared to FairnessGAN.

Table 9: Additional comparisons between baselines using generated data. We compare PairAug and FairnessGAN, where the results are from Ramaswamy et al. (2021). The baselines use the *attractive* attribute as the label and *gender* attribute as the group.

| Method | Acc. | EO Disp. |
|---|---|---|
| PairAug (Ramaswamy et al., 2021) | **80.0** | **0.21** |
| FairnessGAN (Sattigeri et al., 2019) | 71.0 | 0.25 |

### B.5 Other Results on CelebA w.r.t. Demographic Parity

Continuing from Sec. 4.1.1, we perform experiments on the CelebA dataset w.r.t. demographic parity (DP). Table 10 shows the accuracy and fairness performances of the algorithms. Similar to the results in Table 2, Dr-Fairness shows higher fairness than the pre-processing (1:1 data ratio) baselines (i.e., simple sampling, PairAug, and PairAug*) and higher accuracy than the in-processing baselines (i.e., fairness constraint, domain independence, and FairBatch). Here, as the pre-processing baselines are not explicitly designed for DP, their fairness performances in terms of DP are sometimes worse than the original ResNet50 results. Since DP aims to ensure the same positive prediction rates without considering the true labels, the training sometimes needs to overfit on the positive labels for specific groups to improve DP. Thus, simply equalizing the data ratio among groups may not be enough to improve DP compared to the EO case.

Table 10: Performances on the CelebA test set w.r.t. demographic parity (DP) on the binary y (age) and z (gender) scenario. Other settings are identical to Table 2. We mark the best and second best performances among the fairness algorithms with bold and underline, respectively.

| Method | y: age and z: gender | | | |
| --- | --- | --- | --- | --- |
| | Acc. | Bal. Acc. | DP Disp. | Bias Amp. |
| Non-fair | $86.4_{\pm 0.3}$ | $76.4_{\pm 0.2}$ | $0.141_{\pm 0.009}$ | $0.101_{\pm 0.022}$ |
| Simple Sampling | $\mathbf{86.8}_{\pm \mathbf{0.2}}$ | $78.3_{\pm 0.5}$ | $0.132_{\pm 0.010}$ | $0.052_{\pm 0.017}$ |
| PairAug (Ramaswamy et al., 2021) | $85.6_{\pm 0.6}$ | $\mathbf{79.8}_{\pm \mathbf{0.3}}$ | $0.151_{\pm 0.007}$ | $0.030_{\pm 0.006}$ |
| PairAug* (Ramaswamy et al., 2021) | $\underline{86.7}_{\pm 0.2}$ | $79.3_{\pm 0.6}$ | $0.144_{\pm 0.003}$ | $0.053_{\pm 0.008}$ |
| Fair. Const. (Zafar et al., 2017a) | $85.9_{\pm 0.3}$ | $74.3_{\pm 1.3}$ | $0.124_{\pm 0.009}$ | $0.103_{\pm 0.006}$ |
| Dom. Indep. (Wang et al., 2020) | $85.0_{\pm 0.7}$ | $72.4_{\pm 2.1}$ | $0.091_{\pm 0.004}$ | $0.034_{\pm 0.021}$ |
| FairBatch (Roh et al., 2021) | $84.9_{\pm 0.3}$ | $74.0_{\pm 1.5}$ | $\mathbf{0.074}_{\pm \mathbf{0.005}}$ | $\mathbf{-0.034}_{\pm \mathbf{0.004}}$ |
| **Dr-Fairness** | $86.5_{\pm 0.3}$ | $78.2_{\pm 0.1}$ | $\underline{0.088}_{\pm 0.009}$ | $-0.028_{\pm 0.005}$ |

### B.6 Other Results on CelebA with Different Settings

Continuing from Sec. 4.1.1, we perform experiments on the CelebA dataset with different group and label combinations. Tables 11 and 12 show the accuracy and fairness performances on the following two settings: (y, z) = (smiling, gender) and (y, z) = (haircolor, gender). In both cases, we observe consistent results to those in Sec. 4.1.1, where Dr-Fairness achieves high fairness (esp. EO) while not sacrificing accuracy. We note that in these two settings, the bias amplification values of the non-fair baseline are already very small (i.e., good enough), so the fair algorithms may not further improve the bias amplification.

Table 11: Performances on the CelebA test set w.r.t. equalized odds (EO) on the binary y (smiling) and binary z (gender) scenario. We mark the best and second best performances among the fairness algorithms with bold and underline, respectively. Other settings are identical to Table 2.

| Method | z: age | | | |
| --- | --- | --- | --- | --- |
| | Acc. | Bal. Acc. | EO Disp. | Bias Amp. |
| Non-fair | $91.7_{\pm 0.2}$ | $91.7_{\pm 0.2}$ | $0.027_{\pm 0.004}$ | $0.006_{\pm 0.002}$ |
| Simple Sampling | $91.8_{\pm 0.1}$ | $91.8_{\pm 0.1}$ | $0.019_{\pm 0.007}$ | $0.012_{\pm 0.004}$ |
| PairAug (Ramaswamy et al., 2021) | $91.4_{\pm 0.1}$ | $91.4_{\pm 0.1}$ | $0.022_{\pm 0.001}$ | $\mathbf{0.003}_{\pm \mathbf{0.001}}$ |
| PairAug* (Ramaswamy et al., 2021) | $\underline{92.0}_{\pm 0.1}$ | $\underline{92.0}_{\pm 0.1}$ | $0.031_{\pm 0.001}$ | $\mathbf{0.003}_{\pm \mathbf{0.000}}$ |
| Fair. Const. (Zafar et al., 2017b) | $91.3_{\pm 0.5}$ | $91.3_{\pm 0.5}$ | $0.014_{\pm 0.007}$ | $\underline{0.009}_{\pm 0.002}$ |
| Dom. Indep. (Wang et al., 2020) | $91.2_{\pm 0.3}$ | $91.2_{\pm 0.3}$ | $0.014_{\pm 0.000}$ | $\underline{0.009}_{\pm 0.004}$ |
| FairBatch (Roh et al., 2021) | $91.6_{\pm 0.1}$ | $91.6_{\pm 0.1}$ | $\mathbf{0.012}_{\pm \mathbf{0.002}}$ | $0.018_{\pm 0.002}$ |
| **Dr-Fairness** | $\mathbf{92.7}_{\pm \mathbf{0.1}}$ | $\mathbf{92.7}_{\pm \mathbf{0.1}}$ | $\underline{0.013}_{\pm 0.003}$ | $0.009_{\pm 0.004}$ |

Table 12: Performances on the CelebA test set w.r.t. equalized odds (EO) on the non-binary y (haircolor) and binary z (gender) scenario. We mark the best and second best performances among the fairness algorithms with bold and underline, respectively. Other settings are identical to Table 2.

| Method | y: haircolor and z: gender | | | |
| --- | --- | --- | --- | --- |
| | Acc. | Bal. Acc. | EO Disp. | Bias Amp. |
| Non-fair | $83.8_{\pm 0.3}$ | $82.0_{\pm 0.6}$ | $0.399_{\pm 0.019}$ | $0.010_{\pm 0.003}$ |
| Simple Sampling | $83.1_{\pm 0.1}$ | $78.9_{\pm 0.8}$ | $0.322_{\pm 0.006}$ | $0.021_{\pm 0.004}$ |
| PairAug* (Ramaswamy et al., 2021) | $82.7_{\pm 0.9}$ | $\underline{80.0}_{\pm 3.0}$ | $0.374_{\pm 0.029}$ | $\underline{0.018}_{\pm 0.013}$ |
| Dom. Indep. (Wang et al., 2020) | $\underline{83.2}_{\pm 0.2}$ | $79.4_{\pm 0.9}$ | $0.308_{\pm 0.033}$ | $\mathbf{0.011}_{\pm \mathbf{0.006}}$ |
| FairBatch (Roh et al., 2021) | $78.4_{\pm 1.4}$ | $68.7_{\pm 1.3}$ | $\underline{0.085}_{\pm 0.015}$ | $0.072_{\pm 0.022}$ |
| **Dr-Fairness** | $\mathbf{84.5}_{\pm \mathbf{0.2}}$ | $\mathbf{82.3}_{\pm \mathbf{0.7}}$ | $\mathbf{0.029}_{\pm \mathbf{0.009}}$ | $0.023_{\pm 0.009}$ |

### B.7 Accuracy-Fairness Tradeoffs

Continuing from Sec. 4, we visually demonstrate the accuracy-fairness tradeoffs of Dr-Fairness and the baselines on the CelebA and ImageNet People Subtree datasets. Figure 10 shows the accuracy and unfairness performances of the baselines and Dr-Fairness. Here, being on the lower-right indicates higher accuracy and fairness and is thus desirable. In CelebA, Dr-Fairness achieves both better accuracy and fairness performances compared to all the baselines. In ImageNet People Subtree, 1) the baselines Simple Sampling, PairAug, and Dom. Indep. show strictly worse performances than Dr-Fairness, 2) FairBatch achieves higher fairness than ours, but the accuracy degradation is severe, and 3) the non-fair baseline shows high accuracy, but much worse fairness. Thus, we can conclude that Dr-Fairness achieves the best accuracy-fairness tradeoffs in both datasets.

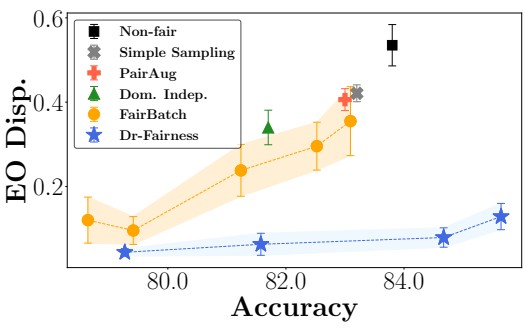
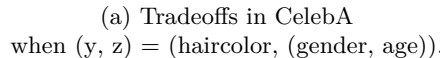
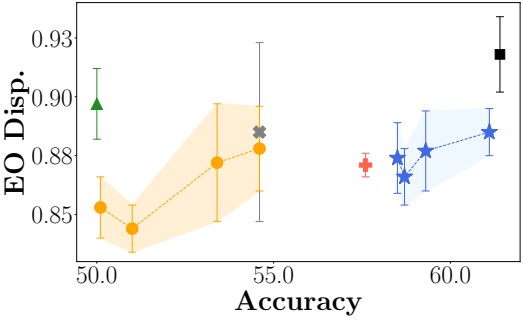

(a) Tradeoffs in CelebA
when (y, z) = (haircolor, (gender, age)).

(b) Tradeoffs in ImageNet People Subtree
when z = (gender, skin color, age).

Figure 10: Accuracy-unfairness graphs to visualize the algorithm performances on the CelebA and ImageNet People Subtree datasets. We note that only Dr-Fairness and FairBatch have accuracy-fairness tuning hyperparameters to produce trade-off curves. The other baselines do not have such tuning knobs, as they use fixed strategies to train models. We thus add the tradeoff curves for Dr-Fairness and FairBatch, while the performances of the other baselines are shown as single points. Being on the lower right is desirable (high accuracy and fairness).

## C  Appendix – Related Work

Continuing from Sec. 2, we discuss more related work.

There are other related studies on 1) fair data reweighing (Li & Liu, 2022; Jiang & Nachum, 2020; Krasanakis et al., 2018), 2) fair augmentation (Chuang & Mroueh, 2021), and 3) fair representations (Shui et al., 2022a).

- The data reweighing techniques (Li & Liu, 2022; Jiang & Nachum, 2020; Krasanakis et al., 2018) are relevant to our data sampling framework, as they keep finding data weights to improve group fairness. Compared to our work, Jiang & Nachum (2020) and Krasanakis et al. (2018) require multiple re-training of the model, and Li & Liu (2022) uses additional assumptions, including the loss function being twice differentiable and strictly convex in the model parameters. Therefore, applying these reweighing techniques when training models on large-scale data may lead to significant training times due to multiple re-trainings or performance degradation due to violations of the assumptions. In comparison, Dr-Fairness works well in large-scale scenarios as in our experiments.

- There is another interesting work called FairMixup (Chuang & Mroueh, 2021), which augments training data for fairness using mixup methods (Zhang et al., 2018b). However, the key difference from ours is that FairMixup only augments the data within the original training data distribution. FairMixup also cannot dynamically adjust sampling ratios from different groups explicitly. In contrast, Dr-Fairness can utilize any additional data, which is not limited to the original training (real) data distribution, and also find the optimal sampling ratios among groups and between real and generated data.

- Recently, a fair representation paper (Shui et al., 2022a) uses the bilevel optimization formulation with the implicit function theorem and shows promising results. Although this work also uses a bilevel formulation, it targets a different problem from ours, where the goal is to map the input feature $X$ into the latent variable $X'$ for fairness. In comparison, we use the bilevel formulation to adjust the ratio among groups and between real and generated data to improve fairness. Specifically, we design the inner objective by explicitly separating the group-wise terms and real/generated data terms to adequately apply the data weights. We note that this inner structure is different from Shui et al. (2022a), where they apply the common outputs of the outer objective to all terms in the inner objective that only considers real data. In addition, when approximating the inverse Hessian matrix resulted by the implicit function theorem (e.g., Eq. 2), Shui et al. (2022a) use the conjugate gradient (CG) method (Rajeswaran et al., 2019), whereas we utilize the identity-matrix approximation (Luketina et al., 2016; Geng et al., 2021). We note that the identity-matrix approximation is known to be much more efficient and may achieve on-par or sometimes better performances compared to the CG method (Lorraine et al., 2020).

