# OpenReview forum: "Dr-Fairness: Dynamic Data Ratio Adjustment for Fair Training on Real and Generated Data"
_TMLR — Accepted by TMLR_

### Review · Reviewer_c8WV · 2023-03-11

**Summary Of Contributions:**

This paper provides an effective training method to promote distributional fairness across different groups (in the notions of EO and DP) by dynamic data ratio adjustment (based on gradients from implicit function theorem and identity matrix approximation) and generated data (from diffusion model with classifier guidance). Experiments on CelebA and ImageNet people subtree show superior fairness or accuracy compared to various types of existing methods.

**Audience:**

Yes

**Broader Impact Concerns:**

The paper contains a Broader Impact Statement that has adequately discussed ethical implications in my opinion.

**Claims And Evidence:**

Yes

**Requested Changes:**

See weaknesses 1 and 2.

**Strengths And Weaknesses:**

Strengths:
1. The paper is very well written and is a pleasure to read.
2. The proposed method is theoretically justified and also effective in practice. The idea of leveraging generated data seems to be novel and reasonable.
3. The empirical evaluation justifies the claim and extensive ablation studies help to explain the mechanism.

Weaknesses:
I didn't find major weaknesses in this submission. Disclaimer: I am not actively working in fairness training so I may miss some context or related work. But there are some minor weaknesses.
1. I found the paper may lack discussion on the training overhead, especially given the extra cost of computing higher-order derivatives in Eqn. (3) and leveraging generative models. I can expect that the training time could be longer than baseline methods, which is acceptable, but this discussion should exist in the paper.
2. It appears to have no reason to put the comparison between FairnessGAN and PairAug in Appendix B.4. Please discuss the reason or move it to the main text and update the findings accordingly.

---

> ### Author Response · Authors · 2023-03-29
> **Response to Reviewer c8WV**
>
> We really appreciate your positive comments and thoughtful suggestions, which helped us improve the manuscript. We made our best efforts to address them.
>
> --------------------
> **Comment 1: Training overhead when computing Equation 3**
>
> As per your great comment, we discuss the training overhead when computing higher-order derivatives in Dr-Fairness. Equation 3 contains two terms: 1) the direct gradient w.r.t. λ and 2) the vector-Jacobian product. Here the second term dominates the computation cost as it performs both the Jacobian computation and vector-Jacobian product. In comparison, the computation cost of the first term is negligible. We also note that the computational cost increases when the model parameter size (i.e., dimension of w) and the number of groups and label classes (i.e., dimension of λ) increase.
>
> Since we currently update λ and μ at every iteration of the training, we can further reduce the computational cost by updating λ and μ less frequently. We added an update-frequency vs training-time graph (Figure 5), which shows that updating λ and μ at every two iterations reduces the training time by half compared to when updating them at each iteration. Interestingly, the overall performance does not degrade significantly as shown in the newly-added update-frequency vs model-performance graphs (Figure 6). Although we have not considered the training overhead issue in our submission, thanks to the reviewer’s suggestion, we believe our new experiment shows the potential of significantly reducing the training overhead without sacrificing the overall performance much. We will add more experiments on real datasets in our final version.
>
> We added this discussion and empirical analysis in our revision (Sections 3.2 and A.2, highlighted in blue).
>
> --------------------
> **Comment 2: Position of Table 8**
>
> We thank you for the suggestion and would like to clarify why we put Table 8 in the appendix. The main role of Table 8 is to explain why we choose PairAug as our main baseline among the fair data augmentation strategies, where Table 8 shows the superiority of PairAug compared to FairnessGAN. We would like to note that Table 8 is not directly related to the comparison with Dr-Fairness. Thus, we believe it is more appropriate to put Table 8 in the appendix in order to keep the experimental parts of the main text more focused on the empirical analysis of Dr-Fairness.
>
> We clarified this point in our revision (Section 4, highlighted in blue).

---

> > ### Comment · Reviewer_c8WV · 2023-04-04
> > **Concerns Addressed**
> >
> > Thank you! My concerns are addressed.

---

### Review · Reviewer_C26t · 2023-03-17

**Summary Of Contributions:**

This paper presents an approach to adjust sampling ratios over different groups and between real and generated data to ensure fairness in the models learned. It proceeds through a bi-level optimization routine that is considering both the accuracy and fairness at the outer level, while adapting sampling weights by taking into account real and generated data instances at the inner level. It also proposes an approximation of the inverse Hessian (in Eq. 2) that aims at reducing computations while maintaining performances. The proposed approach is evaluated on face image classification datasets, with several labels per image (including protected variables as gender or age), with a comparison with other fairness-enforcing approaches. Results show the capacity of the approach to maintain a fairness level comparable to the other approaches and baselines evaluated experimentally, while improving the accuracy (sometime significantly) over the datasets.

**Audience:**

Yes

**Broader Impact Concerns:**

The ethical implications are fine, this is in fact a work that supports approaches for deployment of fairness-aware AI algorithms.

**Claims And Evidence:**

Yes

**Requested Changes:**

Overall, the paper appears great and relevant to me. The main weakness is in the scope of the experiments, which is quite limited to image classification over face images. I get that these datasets are commonly available and fit for the type of experiments presented in the paper, but I think it would be a good improvement if other types of data can be also evaluated with the approach. It would support the fact that the proposed approach has a broad scope of applicability. Medical imaging datasets might be an interesting domain for that purpose, given that it can remain in the topics of image classification, using the same backbone architecture, and having significant fairness considerations, while dealing with types of images significantly different from faces.

Also, the following papers are relevant to the topic but not cited in the manuscript, so I suggest adding them to the literature review.

Changjian Shui, Gezheng Xu, Qi Chen, Jiaqi Li, Charles Ling, Tal Arbel, Boyu Wang, and Christian Gagné. "On learning fairness and accuracy on multiple subgroups." NeurIPS 2022. https://arxiv.org/abs/2210.10837

Joshua K. Lee, Yuheng Bu, Deepta Rajan, Prasanna Sattigeri, Rameswar Panda, Subhro Das, and Gregory W. Wornell. "Fair selective classification via sufficiency." In International Conference on Machine Learning (ICML), pp. 6076-6086. PMLR, 2021.
https://proceedings.mlr.press/v139/lee21b.html


**Strengths And Weaknesses:**

Strong points:
- The proposed approach is presented with a good level of detail, is well justified, and appears sound to me.
- The experiments provide strong evidence that the proposed approach is working as expected.
- The paper is well written and relatively easy to follow.
- The problem of algorithmic fairness is important given the impacts AI can have on people and society and the risk of reproducing the biases found in the datasets in machine learning models.

Weaknesses:
- The paper is tackling a topic in a rather incremental way, it builds upon existing results and proposed approaches with some originality, but it would not qualify it as a foundational paper that would open a new field.
- The proposal is properly justified and follows some intuitions, but is mostly supported through empirical evaluation. Having more developed theoretical justifications would have made the paper even stronger.
- The experiments are limited to image classification benchmarks over faces, the approach is not tested in order contexts that may involve other types of images, or even other modalities (e.g., text).
- Some references are missing (more in next sections).

---

> ### Author Response · Authors · 2023-03-29
> **Response to Reviewer C26t**
>
> We really appreciate your positive comments and thoughtful suggestions, which helped us improve the manuscript. We made our best efforts to address them.
>
> ---------------------
> **Suggestion 1: Experiments using non-facial datasets**
>
> We thank you for the important point and agree that experiments on non-facial datasets are indeed valuable. Related to this issue, we would like to clarify that the images in the ImageNet People Subtree dataset are not necessarily facial images, and instead they are more about people in a general form:
>
> - Many images do not properly show faces. For example, images of the class “black belt” usually show several people dueling with martial arts where their faces cannot be viewed easily. For the class “groom”, there are images containing groups of people facing backwards, so their faces are not even visible. Here the model needs to understand features beyond face information including clothes and background to obtain high classification performance.
> - We would like to also explain that although many existing fairness works in computer vision mainly uses CelebA [1, 2, 3], which may be limited to facial classification, we further analyze the performance of Dr-Fairness on a much more challenging scenario using ImageNet People Subtree (e.g., more than 100 label classes and highly skewed groups) which is more representative of real-world applications.
>
> In addition, we believe using other data like medical images is indeed an interesting direction, as you suggested. One challenge of using medical image datasets in fairness research is the lack of group annotation [4], which is hard to collect or sometimes intentionally removed due to privacy concerns. Although it is not easy to immediately perform an experiment on the medical datasets, we anticipate Dr-Fairness will still work well in the new datasets, as ours shows promising results in complex datasets like ImageNet People Subtree.
>
> Following your comment, we clarified that ImageNet People Subtree represents a non-facial classification task and added the discussion on possible future work (e.g., experiments on medical image datasets) in our revision (Sections 4 and 5, highlighted in blue).
>
> - [1] Ramaswamy et al., “Fair Attribute Classification through Latent Space De-biasing”, CVPR 2021.
> - [2] Choi et al., “Fair Generative Modeling via Weak Supervision”, ICML 2020.
> - [3] Wang et al., “Towards Fairness in Visual Recognition: Effective Strategies for Bias Mitigation”, CVPR 2020.
> - [4] Xu et al., “A Survey of Fairness in Medical Image Analysis: Concepts, Algorithms, Evaluations, and Challenges”, ArXiv 2022.
>
> -------------------------
> **Suggestion 2: Related work**
>
> We appreciate all the papers [1, 2], which are indeed related to our work. Specifically, the first paper [1] focuses on a relevant setting to our work, where it aims to improve fairness for multiple groups each with a limited number of samples, but the paper only focuses on a specific fairness metric called group sufficiency. The second paper [2] also solves an important group fairness issue in selective classification, where a model is allowed to abstain from providing a decision.
>
> We cited these papers in our revision (Section 2, highlighted in blue).
>
> - [1] Shui et al., “On Learning Fairness and Accuracy on Multiple Subgroups”, NeurIPS 2022.
> - [2] Bu et al., “Fair Selective Classification via Sufficiency”, ICML 2021.

---

> > ### Comment · Reviewer_C26t · 2023-04-06
> > **Satisfied with the revision**
> >
> > The answers and revision of the paper are satisfactory, I am confortable with the paper in its current state.

---

### Review · Reviewer_9NWk · 2023-03-17

**Summary Of Contributions:**

This paper investigates the issue of unbalanced data generation, resulting in a bias/fairness concern. The goal is to balance the generated data and control the prediction/fairness performance compared to raw data. To address this problem, the paper proposes a bi-level optimization approach that automatically adjusts the balancing coefficient on subgroups and classes (labels). The proposed method is evaluated on the CelebA/ImageNet dataset, and results demonstrate its effectiveness.

**Audience:**

Yes

**Claims And Evidence:**

Yes

**Requested Changes:**

In general, this paper is well-written. I would suggest authors to address the points in *Are the claims made in the submission supported by accurate, convincing and clear evidence.*



**Strengths And Weaknesses:**

### Review Summary

[*Disclosure: this reviewer has reviewed this paper in other venues*]

This paper addresses an important fairness issue in data-augmentation, specifically in rebalancing the training loss between generated and real data. The authors propose a bi-level objective to automatically balance the weights for sensitive attribute and label balancing. The proposed idea seems sound, and the authors conducted empirical evaluations to support their claims. Therefore I support **acceptance**.

On the negative side, several statements within the paper still require refinements. My comments, based on TMLR guidelines, are as follows.

### Would some individuals in TMLR's audience be interested in the findings of this paper?

Yes. This paper addresses an important fairness issue in data-augmentation, specifically in rebalancing the training loss between generated and real data.

### Are the claims made in the submission supported by accurate, convincing and clear evidence?

Most claims are well-supported. I have few additional comments on some parts

1. Sec 3 Fairness Definitions. For example, in this paper, DP is defined as:
$$P(\hat{Y}=1|Z=z_1)= P(\hat{Y}=1|Z=z_2)$$
A natural question is how could we evaluate the bias such as DP gap, when Z is non-binary? We consider the DP gap for each pair (z_1 and z_2)?

2. Fig 3 seems quite interesting. I was wondering, why not determine the initial ratio with exact inverse Hessian computation? Then the convergence will be much faster.

3.  Remark 4.
> We believe this phenomenon is related to the optimal accuracy-fairness tradeoff, which is known to be determined by the data distribution.  When the performance of a fair algorithm lies on the optimal accuracy-fairness tradeoff, any other algorithm can only achieve either better fairness or better accuracy, but cannot improve both

Agree, there is a lower bound on this.

> However, when the fairness algorithms do not achieve the optimal accuracy-fairness tradeoff in the given data, there is an opportunity to improve the model’s performances toward the optimal tradeoff.

This is not convincing enough. First, each method has its own accuracy-fair trade-off curves. For example, by changing different fair constraints, we could visualize such curves. A fair comparison should visualize such a curve for each method, and demonstrate the proposed method controls a better trade-off. For example, extending Fig 8 to different curves for each method.

4. The Real world example in Fig 1 could better be integrated with real-world generated samples in Fig7. I like the motivating example and I think it is better to consider a better story, for example, including the results in Fig.7.

---

> ### Author Response · Authors · 2023-03-29
> **Response to Reviewer 9NWk**
>
> We really appreciate your positive comments and thoughtful suggestions, which helped us improve the manuscript. We made our best efforts to address them.
>
> ----------------
> **Comment 1: Demographic parity (DP) evaluation for non-binary groups**
>
> We thank you for the question and explain how we could evaluate the DP gap for non-binary groups. In our work, we can measure the maximum positive prediction disparity among groups as follows: DP disp. = max |Pr(**ŷ**=1|**z**=z)−Pr(**ŷ**=1)| for all z. When a model achieves perfect DP (i.e., Pr(**ŷ**=1|**z**=z_1) = Pr(**ŷ**=1|**z**=z_2) for all z_1 and z_2), the DP disparity (gap) becomes zero. This evaluation is also used in FairBatch [1], which measures the DP gap for non-binary groups. We note that the DP gap can also be measured in other ways, including calculating the average of disparities.
>
> We added a pointer from the fairness definition in Section 3 to the measurement methods in Section 4 in our revision (Section 3, highlighted in blue).
>
> [1] Roh et al., FairBatch: Batch Selection for Model Fairness, ICLR 2021.
>
> ----------------
> **Comment 2: Using exact inverse Hessian computation to determine the initial ratio**
>
> We thank you for the suggestion and would like to clarify that the inverse Hessian can be used for computing the *gradients* of ratios (i.e., λ and μ), rather than obtaining an analytical solution for them. Thus, even if we utilize the exact inverse Hessian for computing the gradients (e.g., the gradient of λ) at the first update, we still need initial ratio values (e.g., λ_init) to get the next ratios (e.g., λ_next = λ_init - gradient of λ).
>
> We clarified this point in our revision (Section 3.2, highlighted in blue).
>
> ----------------
> **Comment 3: Accuracy-fairness tradeoff curves**
>
> As per your comment, we updated the tradeoff curves in Figure 10 (previously Figure 8) in our revision (Section B.7, highlighted in blue). We would like to clarify that only Dr-Fairness and FairBatch in Figure 10 have accuracy-fairness tuning hyperparameters to produce trade-off curves. The other baselines, including PairAug and Domain-Independent, do not have such tuning knobs, as they use fixed strategies to train models (e.g., PairAug always generates the same number of images per group). Thus, we added the tradeoff curves for Dr-Fairness and FairBatch in Figure 10, while the performances of the other baselines are shown as single points. As we observe, Dr-Fairness still shows better tradeoffs compared to FairBatch and the other baselines.
>
> ----------------
> **Comment 4: Motivating example in Figure 1**
>
> We appreciate your comment and updated Figure 1 using our real-world generated data in order to strengthen our motivation. In the new figure, we consider a motivating example in ImageNet People Subtree. Specifically, ImageNet People Subtree contains mostly lighter skin color images for anchors. In our new example, we show that the generated data can compensate for the underrepresented groups, i.e., the darker skin color distributions for anchors.
>
> We updated Figure 1 in our revision (Section 1, highlighted in blue).

---

> > ### Comment · Reviewer_9NWk · 2023-03-31
> > **Thank you**
> >
> > Thanks for your rebuttal and revised paper. My minor points have been addressed.

---

### Decision · Action_Editors · 2023-04-16

**Recommendation:** Accept as is

**Comment:**

Reviewers commended that the paper presentation is clear and the proposed method provides solid experimental results for justifying its effectiveness.

Reviewers pointed out the approach is not completely novel which is fine according to TMLR review criteria.

Reviewers also raised questions regarding specific technical points (e.g., how does the proposed approach achieve what kind of fairness, and some specific experimental results). Author feedback addressed most of the questions according to the reviewers.

In final camera ready, I suggest the authors to incorporate the suggestions from the reviewers.

**Audience:**

ML research community working on fairness and imbalanced data.

**Claims And Evidence:**

This paper proposes a bi-level optimisation approach for adaptive sampling of training data from both real and generated datapoint. It aims at automatically adjusting the balancing coefficient on subgroups and classes (labels). The proposed method is evaluated on the CelebA/ImageNet dataset, and results demonstrate its effectiveness.